# Beyond Lazy Training for Over-parameterized Tensor Decomposition

**Xiang Wang**[*]
Duke University
xwang@cs.duke.edu

**Chenwei Wu**[*]
Duke University
cwwu@cs.duke.edu

**Jason D. Lee**
Princeton University
jasonlee@princeton.edu

**Tengyu Ma**
Stanford University
tengyuma@stanford.edu

**Rong Ge**
Duke University
rongge@cs.duke.edu

## Abstract

Over-parametrization is an important technique in training neural networks. In both theory and practice, training a larger network allows the optimization algorithm to avoid bad local optimal solutions. In this paper we study a closely related tensor decomposition problem: given an $l$-th order tensor in $(\mathbb{R}^d)^{\otimes l}$ of rank $r$ (where $r \ll d$), can variants of gradient descent find a rank $m$ decomposition where $m > r$? We show that in a lazy training regime (similar to the NTK regime for neural networks) one needs at least $m = \Omega(d^{l-1})$, while a variant of gradient descent can find an approximate tensor when $m = O^*(r^{2.5l} \log d)$. Our results show that gradient descent on over-parametrized objective could go beyond the lazy training regime and utilize certain low-rank structure in the data.

## 1 Introduction

The success of training neural networks has sparked theoretical research in understanding non-convex optimization. Over-parametrization – using more neurons than the number of training data or than what is necessary for expressivity – is crucial to the success of optimizing neural networks (Livni et al., 2014; Jacot et al., 2018; Mei et al., 2018). The idea of over-parametrization also applies to other related or simplified problems, such as matrix factorization and tensor decomposition, which are of their own interests and also serve as testbeds for analysis techniques of non-convex optimization. We focus on over-parameterized tensor decomposition in this paper (which is closely connected to over-parameterized neural networks (Ge et al., 2018)).

Concretely, given an order-$l$ symmetric tensor $T^*$ in $(\mathbb{R}^d)^{\otimes l}$ with rank $r$, we aim to decompose it into a sum of rank-1 tensors with as few components as possible. Finding the low-rank decomposition with the smallest possible rank $r$ is known to be NP-hard (Hillar and Lim, 2013). The problem becomes easier if we relax the goal to finding a decomposition with $m$ components where $m$ is allowed to be larger than $r$. The natural approach is to optimize the following objective using gradient descent

$$\min_{u_i \in \mathbb{R}^d, c_i \in \mathbb{R}} \left\| \sum_{i=1}^{m} c_i u_i^{\otimes l} - \sum_{i=1}^{r} c_i^* [u_i^*]^{\otimes l} \right\|_F^2. \tag{1}$$

When $m = r$, gradient descent on the objective above will empirically get stuck at a bad local minimum even for orthogonal tensors (Ge et al., 2015). On the other hand, when $m = \Omega(d^{l-1})$,

---

[*]Equal contribution.

gradient descent provably converges to a global minimum near the initialization. This result follows straightforwardly from the Neural Tangent Kernel (NTK) technique (Jacot et al., 2018), which was originally developed to analyze neural network training, and is referred to as the "lazy training" regime because essentially the algorithm is optimizing a convex function near the initialization (Chizat and Bach, 2018a).

The main goal of this paper is to understand whether we can go beyond the lazy training regime for the tensor decomposition problem via better algorithm design and analysis. In other words, we aim to use a much milder over-parametrization than $m = \Omega(d^{l-1})$ and still converge to the global minimum of objective (1). We view the problem as an important first step towards analyzing neural network training beyond the lazy training regime.

We build upon the technical framework of mean-field analysis (Mei et al., 2018), which was developed to analyze overparameterized neural networks. It allows the parameters to move far away from the initialization and therefore has the potential to capture the realistic training regime of neural networks. However, to date, all the provable optimization results with mean-field analysis essentially operate in the infinite or exponential overparameterization regime (Chizat and Bach, 2018b; Wei et al., 2019), and applying these techniques to our problem naively would require $m$ to be exponentially large in $d$, which is even worse than the NTK result. The exponential dependency is *not surprising* because the mean-field analyses in (Chizat and Bach, 2018b; Wei et al., 2019) do not leverage or assume any particular structures of the data so they fail to produce polynomial-time guarantees on the worst-case data. Motivated by identifying problem structure that allows for polynomial-time guarantees, we study the mean-field analysis applied to tensor decomposition.

The main contribution of this paper is to attain nearly dimension-independent over-parametrization for the mean-field analysis in Wei et al. (2019) by leveraging the particular structure of the tensor decomposition problem, and to show that with $m = O^*(r^{2.5l} \log d)$, a modified version of gradient descent on a variant of objective (1) converges to the global minimum and recovers the ground-truth tensor. This is a significant improvement over the NTK requirement of $m = \Omega(d^{l-1})$ and an exponential improvement upon the existing mean-field analysis that requires $m = \exp(d)$. Our analysis shows that unlike the lazy training regime, gradient descent with small initialization and appropriate regularizer can identify the subspace that the ground-truth components lie in, and automatically exploit such structure to reduce the number of necessary components. As shown in Ge et al. (2018), the population-level objective of two-layer networks is a mixture of tensor decomposition objectives with different orders, so our analysis may be extendable to improve the over-parametrization necessary in analysis of two-layer networks.

## 1.1 Related work

**Neural Tangent Kernel** There has been a recent surge of research on connecting neural networks trained via gradient descent with the neural tangent kernel (NTK) (Jacot et al., 2018; Du et al., 2018a,b; Chizat and Bach, 2018a; Allen-Zhu et al., 2018; Arora et al., 2019a,b; Zou and Gu, 2019; Oymak and Soltanolkotabi, 2020). This line of analysis proceeds by coupling the training dynamics of the nonlinear network with the training dynamics of its linearization in a local neighborhood of the initialization, and then analyzing the optimization dynamics of the linearization which is convex.

Though powerful and applicable to any function class including tensor decomposition, NTK is not yet a completely satisfying theory for explaining the success of over-parametrization in deep learning. Neural tangent kernel analysis is essentially dataset independent and requires at least number of neurons $m \geq \frac{n}{d} = d^{l-1}$ to find a global optimum (Zou and Gu, 2019; Daniely, 2019)[2].

**Beyond NTK approach** The gap between linearized models and the full neural network has been established in theory by (Wei et al., 2019; Allen-Zhu and Li, 2019; Yehudai and Shamir, 2019; Ghorbani et al., 2019; Dyer and Gur-Ari, 2019; Woodworth et al., 2020) and observed in practice (Chizat and Bach, 2018a; Arora et al., 2019a; Lee et al., 2019). Higher-order approximations of the gradient dynamics such as Taylorized Training (Bai and Lee, 2019; Bai et al., 2020) and the Neural Tangent Hierarchy (Huang and Yau, 2019) have been recently proposed towards closing this gap. Unlike this paper, existing results mostly try to improve the sample complexity instead of the level of over-parametrization for the NTK approach.

**Mean field approach** For two-layer networks, a series of works used the mean field approach to establish the evolution of the network parameters (Mei et al., 2018; Chizat and Bach, 2018b; Wei et al., 2018; Rotskoff and Vanden-Eijnden, 2018; Sirignano and Spiliopoulos, 2018). In the mean field regime, the parameters move significantly from their initialization, unlike NTK regime, so it is *a priori* possible for the mean field approach to exploit data-dependent structure to utilize fewer neurons. However the current analysis techniques for mean field approach need either exponential in dimension or exponential in time number of neurons to attain small training error and do not exploit any data structure. One of the main contributions of our work is to show that gradient descent can benefit from the low-rank structure in $T^*$.

**Tensor decomposition** Tensor decomposition is in general an NP-hard problem (Hillar and Lim, 2013). There are many algorithms that find the exact decomposition (when $m = r$) under various assumptions. In particular Jenrich's algorithm (Harshman, 1970) works when $r \leq d$ and the components are linearly independent. In our setting, the components may not be linearly independent, this is similar to the overcomplete tensor decomposition problem. Although there are some algorithms for overcomplete tensor decomposition (e.g., Cardoso (1991); Ma et al. (2016)), they require nondegeneracy conditions which we are not assuming. When the number of components $m$ is allowed to be larger than $r$, one can use spectral algorithms to find a decomposition where $m = \Theta(r^{l-1})$. In this paper our focus is to achieve similar guarantees using a direct optimization approach.

**Neural network with polynomial activations** Another model that sits between tensor decomposition and standard ReLU neural network is neural network with polynomial activations. Livni et al. (2013) gave an algorithm for training network with quadratic activations with specific algorithm. Andoni et al. (2014) gave a way to learn degree $l$ polynomials over $d$ variables using $\Omega(d^l)$ neurons, which is similar to the guarantee of (much later) NTK approach.

## 2 Notations

We use $[n]$ as a shorthand for $\{1, 2, \cdots, n\}$. We use $O(\cdot), \Omega(\cdot)$ to hide constant factor dependencies. We use $O^*(\cdot)$ to hide constant factors and also the dependency on accuracy $\epsilon$. We use poly$(\cdot)$ to represent a polynomial on the relevant parameters with constant degree.

**Tensor notations:** We use $\otimes$ as the tensor product (outer product). An $l$-th order $d$-dimensional tensor is defined as an element in space $\mathbb{R}^d \otimes \cdots \otimes \mathbb{R}^d$, succinctly denoted as $(\mathbb{R}^d)^{\otimes l}$. For any $i_1, \cdots, i_l \in [d]$, we use $T_{i_1, \cdots, i_l}$ to refer to the $(i_1, \cdots, i_l)$-th entry of $T \in (\mathbb{R}^d)^{\otimes l}$ with respect to the canonical basis. For a vector $v \in \mathbb{R}^d$, we define $v^{\otimes l}$ as a tensor in $(\mathbb{R}^d)^{\otimes l}$ such that $\left(v^{\otimes l}\right)_{i_1, \cdots, i_l} = v_{i_1} v_{i_2} \cdots v_{i_l}$. A tensor is symmetric if the entry values remain unchanged for any permutation of its indices. We define $\text{vec}(\cdot)$ to be the vectorize operator for tensors, mapping a tensor in $(\mathbb{R}^d)^{\otimes l}$ to a vector in $\mathbb{R}^{d^l}$: $\text{vec}(T)_{(i_1-1)d^{l-1} + (i_2-1)d^{l-2} + \cdots + (i_{l-1}-1)d + i_l} := T_{i_1, i_2, \cdots, i_l}$.

A tensor $T \in (\mathbb{R}^d)^{\otimes l}$ is rank-1 if it can be written as $T = w \cdot v_1 \otimes v_2 \otimes \cdots \otimes v_l$ for some $w \in \mathbb{R}$ and $v_1, \cdots, v_l \in \mathbb{R}^d$, and the rank of a tensor is defined as the minimum integer $k$ such that this tensor equals the sum of $k$ rank-1 tensors.

**Norm and inner product:** We use $\|v\|$ to denote the $\ell_2$ norm of a vector $v$. For $l$-th order tensors $T, T' \in (\mathbb{R}^d)^{\otimes l}$ (vectors and matrices can be viewed as tensors with order 1 and 2, respectively), we define the inner product as $\langle T, T' \rangle := \sum_{i_1, \cdots, i_l \in [d]} T_{i_1, \cdots, i_l} T'_{i_1, \cdots, i_l}$ and the Frobenius norm as $\|T\|_F = \sqrt{\sum_{i_1, \cdots, i_l \in [d]} T^2_{i_1, \cdots, i_l}}$.

## 3 Problem setup and challenges

In this section we discuss the objective for over-parameterized tensor decomposition and explain the challenges in optimizing this objective.

We consider tensor decomposition problems with general order $l \geq 3$. Throughout the paper we consider $l$ as a constant. Specifically, we assume that the ground-truth tensor is $T^*$ of rank at most $r$:

$$T^* := \sum_{i=1}^{r} c_i^* [u_i^*]^{\otimes l},$$

where $\forall i \in [r], c_i^* \in \mathbb{R}$ and $u_i^* \in \mathbb{R}^d$. Without loss of generality, we assume that $\|T^*\|_F = 1$. We focus on the low rank setting where $r$ is much smaller than $d$. Note we don't assume that $u_i^*$'s are linearly independent.

The vanilla over-parameterized model we use consists of $m$ components (where $m \geq r$):

$$T_v := \sum_{i=1}^{m} c_i u_i^{\otimes l},$$

where $\forall i \in [m], c_i \in \mathbb{R}$ and $u_i \in \mathbb{R}^d$. We use $U \in \mathbb{R}^{d \times m}$ to denote the matrix whose $i$-th column is $u_i$, and denote $C \in \mathbb{R}^{m \times m}$ as the diagonal matrix with $C_{i,i} = c_i$. The vanilla loss function we are considering is the square loss:

$$f_v(U, C) = \frac{1}{2} \|T_v - T^*\|_F^2 = \frac{1}{2} \left\| \sum_{i=1}^{m} c_i u_i^{\otimes l} - \sum_{i=1}^{r} c_i^* [u_i^*]^{\otimes l} \right\|_F^2. \tag{2}$$

In other words, we are looking for a rank $m$ approximation to a rank $r$ tensor. When $m = r$, the problem of finding a decomposition is known to be NP-hard. Therefore, our goal is to get a small objective value with small $m$ (which corresponds to the rank of $T_v$). In the following sub-sections, we will see that there are many challenges to directly optimize the vanilla over-parameterized model over the vanilla objective, so we will need to modify the parametrization of the tensor $T_v$ and the optimization algorithm to overcome them.

### 3.1 Challenge 0: lazy training requires immense over-parameterization

We show lazy training requires $\Omega(d^{l-1})$ components to fit a rank-one tensor in the following theorem.

**Theorem 1.** *Suppose the ground truth tensor $T^* = [u^*]^{\otimes l}$, where $u^*$ is uniformly sampled from the unit sphere $\mathbb{S}^{d-1}$. Lazy training (defined as below) requires $\Omega(d^{l-1})$ components to achieve $o(1)$ error in expectation.*

In the lazy training regime, all the $u_i$'s stay very close to the initialization. Assuming the final $u_i'$ is equal to $u_i + \delta_i$, all the higher-order terms in $\delta_i$ can be ignored. Therefore, the model can only capture tensors in the linear subspace $S_U = \text{span}\{P_{sym}\text{vec}(u_i^{\otimes l-1} \otimes \delta_i)\}_{i=1}^{m}$ (here $P_{sym}$ is the projection to the space of vectorized symmetric tensors, $u_i$'s are the initialization and $\delta_i$'s are arbitrary vectors in $\mathbb{R}^d$). The dimension of this subspace is upperbounded by $dm$. Let $W_l$ be the space of all vectorized symmetric tensors in $(\mathbb{R}^d)^{\otimes l}$ (with dimension $\Omega(d^l)$), and $S_U^\perp$ be the subspace of $W_l$ orthogonal to $S_U$. We show that for a random rank-1 tensor $T^*$, it will often have a large projection in $S_U^\perp$ unless $m = \Omega(d^{l-1})$. Basically, the subspace $S_U$ has to cover the whole space $W_l$ to approximate a random rank-1 tensor. The proof of Theorem 1 is in Appendix A.

### 3.2 Challenge 1: zero is a high-order saddle point for vanilla objective

As Chizat and Bach (2018a) pointed out, lazy training regime corresponds to the case where the initialization has large norm. A natural way to get around lazy training is to use a much smaller initialization. However, for the vanilla objective, 0 will be a saddle point of order $l$ on the loss landscape. This makes gradient descent really slow at the beginning. In Section 4, we fix this issue by re-parameterizing the model into a 2-homogeneous model.

### 3.3 Challenge 2: existence of bad local minima far away from 0

It was shown that no bad local minima exist in matrix decomposition problems (Ge et al., 2016). Therefore, (stochastic) gradient descent is guaranteed to find a global minimum. In this section, we show that in contrary tensor decomposition problems with order at least 3 have bad local minima.

**Theorem 2.** *Let $f_v(U, C)$ be as defined in Equation 2. Assume $l \geq 3, d > r \geq 1$ and $m \geq r(l+1) + 1$. There exists a symmetric ground truth tensor $T^*$ with rank at most $r(l+1) + 1$ such that a local minimum with function value $l(l-1)r/4$ exists while the global minimum has function value zero.*

In the construction, we set all the $u_i$'s to be $e_1/m^{1/l}$ so that $T = e_1^{\otimes l}$. We define the ground truth tensor by setting the residual $T - T^*$ to be $\sum_{j=2}^{r+1} e_j^{\otimes 2} \otimes e_1^{\otimes l-2}$ plus its $\binom{l}{2}$ permutations. At this point, the gradient equals zero, so there is no first order change to the function value. Furthermore, we show if any component moves in one of the missing direction $e_j$ for $2 \leq j \leq r+1$, it will incur a second order function value increase. So the tensor can only moves along $e_1$ direction, which cannot further decrease the function value because $e_1^{\otimes l}$ is orthogonal with the residual. Note this is a bad local min but not a strict bad local min because we can shrink one component to zero and meanwhile increase another component so that the tensor does not change. When we have a zero component (it's a saddle point), we can add a missing direction to decrease the function value.

In Appendix B, we prove Theorem 2 and also construct a bad local minimum for 2-homogeneous model defined in Section 4. To escape these spurious local minima, our algorithm re-initializes one component after a fixed number of iterations.

# 4 Algorithms and main results

In this section, we introduce our main algorithm, a modified version of gradient descent on a non-convex objective, and state our main results.

To address the high-order saddle point issue in Section 3.2, we introduce a new variant of the parameterized models.

$$T := \sum_{i=1}^{m} a_i c_i^{l-2} u_i^{\otimes l},$$

where $\forall i \in [m], a_i \in \{-1, 1\}, c_i = \frac{1}{\|u_i\|}$ and $u_i \in \mathbb{R}^d$.

Note that since $u_i^{\otimes l}$ is homogeneous, there is a redundancy in the vanilla parametrization between the coefficient and the norm of $\|u\|$. Here we do the rescaling to make sure that the model $T$ is a 2-homogeneous function of $u_i$'s. Using the new formulation of $T$, 0 will no longer be a high order saddle point.

Recall that we use $U \in \mathbb{R}^{d \times m}$ to denote the matrix whose $i$-th column is $u_i$. We use $C, A \in \mathbb{R}^{m \times m}$ to denote the diagonal matrices with $C_{ii} = c_i, A_{ii} = a_i$. The loss function we are considering is the square loss plus a regularization term:

$$f(U, C, \hat{C}, A) \triangleq \frac{1}{2} \left\| \sum_{i=1}^{m} a_i c_i^{l-2} u_i^{\otimes l} - T^* \right\|_F^2 + \lambda \sum_{i=1}^{m} \hat{c}_i^{l-2} \|u_i\|^l,$$

where $\forall i \in [m], \hat{c}_i \in \mathbb{R}^+$ and we use $\hat{C} \in \mathbb{R}^{m \times m}$ to denote the diagonal matrix with $\hat{C}_{ii} = \hat{c}_i$. For simplicity, we use $\bar{C}$ to denote the tuple $(C, \hat{C}, A)$. Therefore, we can write $f(U, C, \hat{C}, A)$ as $f(U, \bar{C})$.

The algorithm contains $K$ epochs, where each epoch includes $H$ iterations. At the initialization, we independently sample each $u_i$ from $\delta \text{Unif}(\mathbb{S}^{d-1})$, where the radius $\delta$ will be set to be $\text{poly}(\epsilon, 1/d)$.

Denote the subspace of $\text{span}\{u_i^*\}$ as $S$ and its orthogonal subspace in $\mathbb{R}^d$ as $B$. Let $P_S, P_B$ be the projection matrices onto subspace $S$ and $B$, respectively. Since the components of the ground-truth tensor lies in the subspace $S$, ideally we want to make sure that the components of tensor $T$ lies in the same subspace $S$. We also want to make sure $c_i$ roughly equals $1/\|P_S u_i\|$ to ensure the improvement in $S$ subspace is large enough. However, the algorithm does not know the subspace $S$. We address this problem using the observation that $\|P_S u_i\| \approx \frac{\sqrt{r}}{\sqrt{d}} \|u_i\|$ at initialization; and $\|P_S u_i\| \approx \|u_i\|$ if norm of $u_i$ is large, but its projection in $B$ has not grown larger. In our algorithm we introduce a "scalar mode switch" step between these two regimes by the separation between $C$ and $\hat{C}$: For the $i$-th component, the coefficients $c_i$ and $\hat{c}_i$ are initialized as $\sqrt{d(m+K)}/\|u_i\|$ and $1/\|u_i\|$, respectively,

and we reduce $c_i$ by a factor of $\sqrt{d(m+K)}$ ($c_i$ will be equal to $\hat{c}_i$ afterwards) when $\|u_i\|$ exceeds $2\sqrt{m+K}\delta$ for the first time. For each $i \in [m]$, $a_i$ is i.i.d. sampled from Unif$\{1, -1\}$.

We also re-initialize one component at the beginning of each epoch. At each iteration, we first update $U$ by gradient descent: $U' \leftarrow U - \eta \nabla_U f(U, \bar{C})$. Note that when taking the gradient over $U$, we treat $c_i$'s and $\hat{c}_i$'s as constants. Then we update each $c_i$ and $\hat{c}_i$ using the updated value of $u_i$ to preserve 2-homogeneity, i.e., $c_i' = \frac{\|u_i\|}{\|u_i'\|} c_i$ and $\hat{c}_i' = \frac{\|u_i\|}{\|u_i'\|} \hat{c}_i$. We fix $a_i$'s during the algorithm except for the initialization and re-initialization steps.

The pseudocode is given in Algorithm 1. Using this variant of gradient descent, we can recover the ground truth tensor $T^*$ with high probability using only $O\left(\frac{r^{2.5l}}{\epsilon^5} \log(d/\epsilon)\right)$ number of components. The formal theorem is stated below.

---

**Algorithm 1** Variant of Gradient Descent for Tensor Decomposition

---

**Input:** number of epochs $K$, number of iterations in one epoch $H$, initialization size $\delta$, step size $\eta$.
For each $i \in [m]$, initialize $u_i$ i.i.d. from $\delta \text{Unif}(\mathbb{S}^{d-1})$; initialize $a_i$ i.i.d. from Unif$\{1, -1\}$; initialize $c_i$ as $\frac{\sqrt{d(m+K)}}{\|u_i\|}$ and $\hat{c}_i$ as $\frac{1}{\|u_i\|}$.
**for** epoch $k := 1$ to $K$ **do**
&emsp;Let $u_j$ be any vector with the smallest $\ell_2$ norm among all columns of $U$.
&emsp;Re-initialize $u_j$ from $\delta \text{Unif}(\mathbb{S}^{d-1})$, re-initialize $a_j$ from Unif$\{1, -1\}$ and set $c_j = \frac{\sqrt{d(m+K)}}{\|u_j\|}, \hat{c}_j = \frac{1}{\|u_j\|}$.

&emsp;**for** iteration $t := 1$ to $H$ **do**
&emsp;&emsp;$U' \leftarrow U - \eta \nabla_U f(U, \bar{C})$.
&emsp;&emsp;**for** $i := 1$ to $m$ **do**
&emsp;&emsp;&emsp;$c_i' \leftarrow \frac{\|u_i\|}{\|u_i'\|} c_i$; $\hat{c}_i' \leftarrow \frac{\|u_i\|}{\|u_i'\|} \hat{c}_i$.
&emsp;&emsp;&emsp;**if** $\|u_i\| \leq 2\sqrt{m+K}\delta < \|u_i'\|$ holds for the first time since it was (re)-initialized
**then**
$$c_i' \leftarrow \frac{c_i'}{\sqrt{d(m+K)}}. \qquad \qquad \triangleright \text{Scalar Mode Switch}$$
&emsp;&emsp;$U \leftarrow U', C \leftarrow C', \hat{C} \leftarrow \hat{C}$.
**Output:** $T := \sum_{i=1}^m a_i c_i^{l-2} u_i^{\otimes l}$.

---

**Theorem 3.** *Given any target accuracy $\epsilon > 0$, there exists $m = O\left(\frac{r^{2.5l}}{\epsilon^5} \log(d/\epsilon)\right), \lambda = O\left(\frac{\epsilon}{r^{0.5l}}\right), \delta = poly(\epsilon, 1/d), \eta = poly(\epsilon, 1/d), H = poly(1/\epsilon, d)$ such that with probability at least $0.99$, our algorithm finds a tensor $T$ satisfying*

$$\|T - T^*\|_F \leq \epsilon,$$

*within $K = O\left(\frac{r^{2l}}{\epsilon^4} \log(d/\epsilon)\right)$ epochs.*

## 5 Summary of our techniques

In this section, we discuss the high-level ideas that we need to prove Theorem 3. The full proof is deferred into Appendix C.

Generally, doing gradient descent never increases the objective value (though this is not obvious for our algorithm as it is slightly different in handling the normalization $c_i, \hat{c}_i$'s). Our main concern is to address Challenge 2, namely, the algorithm might get stuck at a bad local minimum. We will show that this cannot happen with the re-initialization procedure.

More precisely, we rely on the following main lemma to show that as long as the objective is large, there is at least a constant probability to improve the objective within one epoch.

**Lemma 1.** *In the setting of Theorem 3, let $(U_0', \bar{C}_0')$ and $(U_H, \bar{C}_H)$ be the parameters at the beginning of an epoch and the parameters at the end of the same epoch. Assume $\|T_0' - T^*\|_F \geq \epsilon$, where $T_0'$ is*

*tensor with parameters $(U_0', \bar{C}_0')$. Then with probability at least $1/6$, we have*

$$f(U_H, \bar{C}_H) - f(U_0', \bar{C}_0') = -\Omega\left(\frac{\epsilon^4}{r^{2l}\log(d/\epsilon)}\right).$$

We complement this lemma by showing that even if an epoch does not decrease the objective, it will not overly increase the objective.

**Lemma 2.** *In the setting of Theorem 3, let $(U_0', \bar{C}_0')$ and $(U_H, \bar{C}_H)$ be the parameters at the beginning of an epoch and the parameters at the end of same epoch. Assume $f(U_0', \bar{C}_0') \geq \epsilon^2$, where $\epsilon$ is the target accuracy in Theorem 3. Then, we have $f(U_H, \bar{C}_H) - f(U_0', \bar{C}_0') = O(\frac{1}{\lambda m})$.*

From these two lemmas, we know that in each epoch, the loss function can decrease by $\Omega\left(\frac{\epsilon^4}{r^{2l}\log(d/\epsilon)}\right)$ with probability at least $\frac{1}{6}$, and even if we fail to decrease the function value, the increase of function value is at most $O\left(\frac{1}{\lambda m}\right)$. By our choice of parameters in Theorem 3, $m = \Theta\left(\frac{r^{2.5l}}{\epsilon^5}\log(d/\epsilon)\right), \lambda = \Theta\left(\frac{\epsilon}{r^{0.5l}}\right)$ and then $O(\frac{1}{\lambda m}) = O(\frac{\epsilon^4}{r^{2l}\log(d/\epsilon)})$. Choosing a large constant factor in $m$, we can ensure that the function value decrease will dominate the increase. This allows us to prove Theorem 3.

In the next two subsections, we will discuss how to prove Lemma 1 and Lemma 2, respectively.

## 5.1 Proof sketch for Lemma 2 - upper bound on function increase

To prove the increase of $f$ is bounded in one epoch, we identify all the possible ways that the loss can increase and upper bound each of them. We first show that a normal step (without scalar mode switch) of the algorithm will not increase the objective function

**Lemma 3.** *In the setting of Theorem 3, let $(U, \bar{C})$ be the parameters at the beginning of one iteration and let $U', \bar{C}'$ be the updated parameters (before potential scalar mode switch). Assuming $f(U, \bar{C}) \leq 10$, we have $f(U', \bar{C}') - f(U, \bar{C}) \leq -\frac{\eta}{l}\left\|\nabla_U f(U, \bar{C})\right\|_F^2$.*

Note that we treat $C$ and $\hat{C}$ as constants when taking gradient with respect to $U$ and then update $C$ and $\hat{C}$ according to the updated value of $U$, so this lemma does not directly follow from standard optimization theory. The gradient descent on $U$ decreases the function value when the step size is small enough while updating $C, \hat{C}$ can potentially increase the function value. In order to show that overall the function value decreases, we need to bound the function value increase due to updating $C, \hat{C}$. We are able to do this because of the special regularizer we choose. In particular, our regularizer guarantees that the change introduced by updating $C$ and $\hat{C}$ is proportional to the change of the gradient step, and is smaller in scale. Therefore we maintain the decrease in the gradient step.

Since we already know that the function value cannot increase in a normal iteration (before potential scalar mode switch), the only causes of the function value increase are the re-initialization or scalar mode switches. According to the algorithm, we only switch the scalar mode when the norm of a component reaches $2\sqrt{m+K}\delta$ for the first time, so the number of scalar mode switches in each epoch is at most $m$. Choosing $\delta$ to be small enough, the effects of scalar mode switches should be negligible. In the re-intialization, we remove the component with smallest $\ell_2$ norm, which can increase the function value by at most $O(\frac{1}{\lambda m})$. This is proved in Lemma 4.

**Lemma 4.** *In the setting of Theorem 3, let $(U_0', \bar{C}_0')$ and $(U_0, \bar{C}_0)$ be the parameters before and after the reinitialization step, respectively. Assume $f(U_0', \bar{C}_0') \geq \epsilon^2$, where $\epsilon$ comes from Theorem 3. Then, we have $f(U_0, \bar{C}_0) - f(U_0', \bar{C}_0') = O(\frac{1}{\lambda m})$.*

In the proof, we can show the function value is at most a constant and then $\sum_{i=1}^m \|u_i\|^2 = O(1/\lambda)$ due to the regularizer. Since we choose the reinitialized component $u$ as one of the component with smallest $\ell_2$ norm, we know $\|u\|^2 = O(\frac{1}{\lambda m})$. This then allows us to bound the function value change from reinitialization by $O(\frac{1}{\lambda m})$. Lemma 2 follows from Lemma 3 and Lemma 4.

## 5.2 Proof sketch for Lemma 1 - escaping local minima

In this section, we will show how we can escape local minima by re-initialization. Intuitively, we will show that when a component is randomly re-initialized, it has a positive probability of having a good

correlation with the current residual $T - T^*$. However, there is a major obstacle here: because the component is re-initialized in the full $d$-dimensional space, the correlation of this new component with $T - T^*$ is going to be of the order $d^{-l/2}$. If every epoch can only improve the objective function by $d^{-l/2}$ we would need a much larger number of epochs and components.

We solve this problem by observing that both $T$ and $T^*$ are almost entirely in the subspace $S$. If we only care about the projection in $S$, the random component will have a correlation of $r^{-l/2}$ with the residual $T - T^*$. We will show that such a correlation will keep increasing until the norm of the new component is large enough, therefore decreasing the objective function.

First of all, we need to show that the influence coming from the subspace $B$ (the orthogonal subspace of the span of $\{u_i^*\}$) is small enough so that it can be ignored.

**Lemma 5.** *In the setting of Theorem 3, we have $\|P_B U\|_F^2 \leq (m + K)\delta^2$ throughout the algorithm.*

We prove Lemma 5 by showing the norm of $P_B U$ only increases at the (re-)initializations, so it will stay small throughout this algorithm. This lemma is also the motivation of our algorithm, i.e., we treat $C$ and $\hat{C}$ as constants when taking the gradient so that the gradient of $U$ will never have negative correlation with $P_B U$.

Now let us focus on the subspace $S$. We denote the re-initialized vector at $t$-th step as $u_t$, and its sign as $a \in \{\pm 1\}$, and we will take a look at the change of $P_S u_t$. Our analysis focuses on the correlation between $P_S u_t$ and the residual tensor: $\langle (P_{S^{\otimes l}} T_t - T^*), a(\overline{P_S u_t}^{\otimes l}) \rangle$. Here $\overline{P_S u_t}$ is the normalized version $P_S u_t$. We will show that the norm of $u_t$ will blow up exponentially if this correlation is significantly negative at every iteration.

Towards this goal, first we will show that the initial point $P_S u_0$ has a large negative correlation with the residual. We lower bound this correlation by anti-concentration of Gaussian polynomials:

**Lemma 6.** *Suppose the residual at the beginning of one epoch is $T_0' - T^*$. Suppose $ac_0^{l-2} u_0^{\otimes l}$ is the reinitialized component. With probability at least $1/5$,*

$$\left\langle P_{S^{\otimes l}} T_0' - T^*, a\overline{P_S u_0}^{\otimes l} \right\rangle \leq -\Omega\left(\frac{1}{r^{0.5l}}\right) \left\| P_{S^{\otimes l}} T_0' - P_{S^{\otimes l}} T^* \right\|_F,$$

*where $\overline{P_S u_0} = P_S u_0 / \|P_S u_0\|$.*

Our next step argues that if this negative correlation is large in every step, then the norm of $u_t$ blows up exponentially:

**Lemma 7.** *In the setting of Theorem 3, within one epoch, let $T_0$ be the tensor after the reinitilization and let $T_\tau$ be the tensor at the end of the $\tau$-th iteration. Assume $\|P_S u_0\| \geq \Omega(\delta/\sqrt{d})$. For any $t \geq 1$, as long as $\left\langle P_{S^{\otimes l}} T_\tau - T^*, a\overline{P_S u_\tau}^{\otimes l} \right\rangle \leq -\Omega\left(\frac{\epsilon}{r^{0.5l}}\right)$ for all $t - 1 \geq \tau \geq 0$, we have*

$$\|P_S u_t\|^2 \geq \left(1 + \Omega\left(\frac{\eta\epsilon}{r^{0.5l}}\right)\right)^t \|P_S u_0\|^2.$$

Therefore the final step is to show that $P_S u_t$ always have a large negative correlation with $T_t - T^*$, unless the function value has already decreased. The difficulty here is that both the current reinitialized component $u_t$ and other components are moving, therefore $T_t$ is also changing.

We can bound the change of the correlation by separating it into two terms, which are the change of the re-initialized component and the change of the residual:

$$\left\langle P_{S^{\otimes l}} T_t - T^*, a\overline{P_S u_t}^{\otimes l} \right\rangle - \left\langle P_{S^{\otimes l}} T_0 - T^*, a\overline{P_S u_0}^{\otimes l} \right\rangle$$

$$\leq \sum_{\tau=1}^{t} \left( \left\langle P_{S^{\otimes l}} T_{\tau-1} - T^*, \overline{P_S u_\tau}^{\otimes l} \right\rangle - \left\langle P_{S^{\otimes l}} T_{\tau-1} - T^*, \overline{P_S u_{\tau-1}}^{\otimes l} \right\rangle \right) + \sum_{\tau=1}^{t} \|T_\tau - T_{\tau-1}\|_F.$$

The change of the re-initialized component has a small effect on the correlation because the change in $S$ subspace can only improve the correlation, and the influence of the $B$ subspace can be bounded. This is formally proved in the following lemma.

**Lemma 8.** *In the setting of Theorem 3, suppose at the beginning of one iteration, the tensor $T$ has parameters $(U, \bar{C})$. Suppose $u$ is one column vector in $U$ with $\|P_S u\| = \Omega(\frac{\delta}{\sqrt{d}})$ and $u' = u - \eta \nabla_u f(U, \bar{C})$. We have*

$$\left\langle P_{S^{\otimes l}} T - T^*, a \overline{P_S u'}^{\otimes l} \right\rangle \leq \left\langle P_{S^{\otimes l}} T - P_{S^{\otimes l}} T^*, a \overline{P_S u}^{\otimes l} \right\rangle + \eta \delta poly(d),$$

*where $poly(d)$ does not hide any dependency on $\eta, \delta$.*

Therefore, the only way to change the residual term by a lot must be changing the tensor $T$, and the accumulated change of $T$ is strongly correlated with the decrease of $f$. This is similar to the technique of bounding the function value decrease in Wei et al. (2019). The connection between them are formalized in the following lemma:

**Lemma 9.** *In the same setting of Lemma 7, within one epoch, let $(U_0, \bar{C}_0)$ be the parameters after the reinitialization step and let $(U_H, \bar{C}_H)$ be the parameters at the end of this epoch. We have*

$$\sum_{\tau=1}^{H} \|T_\tau - T_{\tau-1}\|_F \leq O\left(\sqrt{\frac{\eta H}{\lambda}}\right) \sqrt{f(U_0, \bar{C}_0) - f(U_H, \bar{C}_H) + \delta^2 poly(d)} + \delta^2 poly(d),$$

*where $poly(d)$ does not hide dependency on $\delta$.*

Intuitively, Lemma 9 is true because a large accumulated change of $T$ indicates large gradients along the trajectory, which suggests a large decrease in the function value. In fact, we choose the parameters such that $\lambda = \Theta(\frac{\epsilon}{r^{0.5l}}), \eta H = \Theta(\frac{r^{0.5l}}{\epsilon} \log(d/\epsilon))$. If the accumulated change of $T$ is larger than $\Omega(\frac{\epsilon}{r^{0.5l}})$, the function value decreases by at least $\Omega(\frac{\epsilon^4}{r^{2l} \log(d/\epsilon)})$, as stated in Lemma 1.

Combining all the steps, we show that either the function value has already decreased (by Lemma 9), or the correlation remains negative and the norm $\|P_S u_t\|$ blows up exponentially (by Lemma 7). The norm cannot grow exponentially because of the regularizer, so the function value must eventually decrease. This finishes the proof of Lemma 1.

# 6    Conclusion

In this paper we show that for an over-parameterized tensor decomposition problem, a variant of gradient descent can learn a rank $r$ tensor using $O^*(r^{2.5l} \log(d/\epsilon))$ components. The result shows that gradient-based methods are capable of leveraging low-rank structure in the input data to achieve lower level of over-parametrization. There are still many open problems, in particular extending our result to a mixture of tensors of different orders which would have implications for two-layer neural network with ReLU activations. We hope this serves as a first step towards understanding what structures can help gradient descent to learn efficient representations.

## Broader Impact

This work does not present any foreseeable societal consequence.

## Acknowledgements

RG acknowledges support from NSF CCF1704656, NSF CCF-1845171 (CAREER), NSF CCF-1934964 (TRIPODS), NSF-Simons Research Collaborations on the Mathematical and Scientific Foundations of Deep Learning, a Sloan Fellowship, and a Google Faculty Research Award.

JDL acknowledges support of the ARO under MURI Award W911NF-11-1-0303, the Sloan Research Fellowship, and NSF CCF 2002272. This is part of the collaboration between US DOD, UK MOD and UK Engineering and Physical Research Council (EPSRC) under the Multidisciplinary University Research Initiative.

TM acknowledges support of Google Faculty Award. The work is also partially supported by SDSI and SAIL at Stanford.

## Footnotes

[2]Here $n$ is the number of samples which is effectively $\Theta(d^l)$ in our setting.

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
