[Supplementary Material]

**Notations**

Besides the notations defined in Section 2, we also use the following notations in the proofs.

We use $\odot$ for the Khatri-Rao product. We denote $e_i$ as the $i$-th basis vector in $\mathbb{R}^d$.

We define $\text{mat}(\cdot)$ to be the matrixize operator for tensors, mapping a tensor in $(\mathbb{R}^d)^{\otimes l}$ to a matrix in $\mathbb{R} \times \mathbb{R}^{d^{l-1}}$: $\text{mat}(T)_{i_1,(i_2-1)d^{l-2}+\cdots+(i_{l-1}-1)d+i_l} := T_{i_1,i_2,\cdots,i_l}$ for any $i_1,i_2,\cdots,i_l \in [d]$.

We view a tensor $T \in (\mathbb{R}^d)^{\otimes l}$ as a multilinear form. For matrices $M_1 \in \mathbb{R}^{d \times k_1}, \cdots, M_l \in \mathbb{R}^{d \times k_l}$, the tensor $T(M_1, M_2, \cdots, M_l) \in \mathbb{R}^{k_1 \times k_2 \times \cdots \times k_l}$ is defined such that

$$T(M_1, M_2, \cdots, M_l)_{j_1,\cdots,j_l} := \sum_{i_1,\cdots,i_l \in [d]} T_{i_1,\cdots,i_l}(M_1)_{i_1,j_1} \cdots (M_l)_{i_l,j_l},$$

for any $j_1 \in [k_1], \cdots, j_l \in [k_l]$. For notation simplicity, we use $T(M^{\otimes k}, M_{k+1}, \cdots, M_l)$ to denote $T(M, M, \cdots, M, M_{k+1}, \cdots, M_l)$. In particular, for any $v \in \mathbb{R}^d$, $T(v^{\otimes l})$ is a scalar equals to $\langle T, v^{\otimes l} \rangle = \sum_{i_1,\cdots,i_l \in [d]} T_{i_1,\cdots,i_l} v_{i_1} v_{i_2} \cdots v_{i_l}$.

# A  Lower Bound for the Number of Components Needed for Kernels

In this section, we will prove that a lazy training model requires $\Omega(d^{l-1})$ components to fit a random rank-one tensor with $o(1)$ loss. Recall Theorem 1 as follows.

**Theorem 1.** *Suppose the ground truth tensor $T^* = [u^*]^{\otimes l}$, where $u^*$ is uniformly sampled from the unit sphere $\mathbb{S}^{d-1}$. Lazy training (defined as below) requires $\Omega(d^{l-1})$ components to achieve $o(1)$ error in expectation.*

Recall that in our definition, a lazy training model can only capture tensors in the linear subspace $S_U = \text{span}\{P_{sym}\text{vec}(u_i^{\otimes l-1} \otimes \delta_i)\}_{i=1}^m$ (here $P_{sym}$ is the projection to the space of vectorized symmetric tensors, $\delta_i$'s are arbitrary vectors in $\mathbb{R}^d$). The dimension of this subspace is upperbounded by $dm$. Let $W_l$ be the space of all vectorized symmetric tensors in $(\mathbb{R}^d)^{\otimes l}$, and $S_U^\perp$ be the subspace of $W_l$ orthogonal to $S_U$. We only need to show that for a random rank-one tensor, in expectation its projection on the orthogonal subspace $S_U^\perp$ is at least a constant unless $m = \Omega(d^{l-1})$. In the following lemma, we first lower bound the projection of the ground truth tensor on a fixed direction. The proof of Lemma 10 is deferred into Section A.2.

**Lemma 10.** *Let $u \in \mathbb{R}^d$ be a vector sampled uniformly on the unit sphere $\mathbb{S}^{d-1}$. For any vectorized symmetric $l$-th order tensor $b \in \mathbb{R}^{d^l}$ with unit $\ell_2$ norm, we have*

$$b^\top \mathbb{E}[vec(u^{\otimes l})vec(u^{\otimes l})^\top]b \geq \frac{\Gamma\left(\frac{d}{2}\right)}{2^l \Gamma\left(l + \frac{d}{2}\right)} l!,$$

*where $\Gamma(\cdot)$ is the Gamma function.*

Next, we lower bound the projection of $\text{vec}(T^*)$ on subspace $S_U^\perp$ by summation up the projections on the subspace bases, each of which can be bounded by Lemma 10. We give the proof of Theorem 1 as follows.

**Proof of Theorem 1.** Recall that $W_l$ is the space of all vectorized symmetric tensors in $(\mathbb{R}^d)^{\otimes l}$. Due to the symmetry, the dimension of $W_l$ is $\binom{d+l-1}{l}$. Since the dimension of $S_U$ is at most $dm$, we know that the dimension of $S_U^\perp$ is at least $\binom{d+l-1}{l} - dm$. Assuming $S_U^\perp$ is an $\bar{m}$-dimensional space, we have $\bar{m} \geq \binom{d+l-1}{l} - dm \geq \frac{d^l}{l!} - dm$. Let $\{e_1, \cdots, e_{\bar{m}}\}$ be a set of orthonormal bases of $S_U^\perp$, and $\Pi_U^\perp$ be the projection matrix from $\mathbb{R}^{d^l}$ onto $S_U^\perp$, then we know that the smallest possible error that we can get given $U$ is

$$\frac{1}{2}\mathbb{E}_{u^*}\left[\left\|\Pi_U^\perp\text{vec}(T^*)\right\|_F^2\right] = \frac{1}{2}\mathbb{E}_{u^*}\left[\sum_{i=1}^{\bar{m}} \langle \text{vec}(T^*), e_i \rangle^2\right] = \frac{1}{2}\sum_{i=1}^{\bar{m}} \mathbb{E}_{u^*}\left[\langle \text{vec}(T^*), e_i \rangle^2\right],$$

where the expectation is taken over $u^* \sim \text{Unif}(\mathbb{S}^{d-1})$.

By Lemma 10, we know that for any $i \in [m]$,

$$\mathbb{E}_{u^*}\left[\langle \text{vec}(T^*), e_i \rangle^2\right] = e_i^\top \mathbb{E}_{u^*}[(\text{vec}([u^*]^{\otimes l})\text{vec}([u^*]^{\otimes l})^\top]e_i$$

$$\geq \frac{\Gamma\left(\frac{d}{2}\right)}{2^l \Gamma\left(l + \frac{d}{2}\right)} l! \geq \mu \frac{l!}{d^l},$$

where $\mu$ is a positive constant only related to $l$.

Therefore,

$$\frac{1}{2}\mathbb{E}_{u^*}\left[\left\|\Pi_{\hat{U}}^\perp T^*\right\|_F^2\right] = \frac{1}{2}\sum_{i=1}^{\bar{m}}\mathbb{E}_{u^*}\left[\langle \text{vec}(T^*), e_i \rangle^2\right] \geq \left(\frac{d^l}{l!} - dm\right)\frac{\mu l!}{2d^l} = \frac{\mu}{2} - \frac{\mu l!}{2} \cdot \frac{m}{d^{l-1}}.$$

Note that we assume $l$ is a constant. If $m = o(d^{l-1})$, i.e., $\frac{m}{d^{l-1}} = o(1)$, then the expectation of the smallest possible error is at least some constant. Thus, if we want the error to be $o(1)$, we must have $m = \Omega(d^{l-1})$. This finishes the proof of Theorem 1. □

## A.1 Numerical verification of the lower bound

In this section, we plot the projection of the ground truth tensor on the orthogonal subspace $\mathbb{E}_{u^*}\left\|\Pi_{\hat{U}}^\perp T^*\right\|_F^2$ under different dimension $d$ and number of components $m$. For convenience, we only plot the lower bound for the projection that is $\left(\binom{d+l-1}{l} - dm\right)\frac{\Gamma\left(\frac{d}{2}\right)}{2^l \Gamma\left(l+\frac{d}{2}\right)}l!$ as we derived previously.

Figure 1 shows that under different dimensions, $\mathbb{E}_{u^*}\left\|\Pi_{\hat{U}}^\perp T^*\right\|_F^2$ is at least a constant until $\log_d m$ gets close to $l - 1 = 3$. As dimension $d$ increases, the threshold when the orthogonal projection significantly drops becomes closer to 3.

Figure 1: The projection of the ground truth tensor on the orthogonal subspace when $l = 4$.

## A.2 Proofs of technical lemmas

To prove Lemma 10, we need another technical lemma, which is stated and proved below.

**Lemma 11.** *Let $u \in \mathbb{R}^d$ be a standard normal vector. For any vectorized symmetric $l$-th order tensor $b \in \mathbb{R}^{d^l}$ with unit norm, we have*

$$b^\top \mathbb{E}[vec(u^{\otimes l})vec(u^{\otimes l})^\top]b \geq l!.$$

**Proof of Lemma 11.** First we define the notion of symmetry: For a vector $x \in \mathbb{R}^{d^B}$, where $B \in \mathbb{N}^*$, if for all permutation $\sigma$ of $[d]$, and for all $i_1, \cdots, i_B \in [d]$, $x_{i_1 i_2 \cdots i_B} = x_{\sigma(i_1)\sigma(i_2)\cdots\sigma(i_B)}$, then we say vector $x$ is symmetric.

Besides, for a vector $v \in \mathbb{R}^{d^l}$, we use $v_{i_1, i_2, \cdots, i_l}$ to refer to $\mathrm{Tensor}(v)_{i_1, i_2, \cdots, i_l}$ where Tensor is the inverse translation of vec, i.e., Tensor translates a $\mathbb{R}^{d^l}$ vector back to a $\mathbb{R}^{d^{\otimes l}}$ tensor. In other words, we use $v_{i_1, i_2, \cdots, i_l}$ to refer to the entry $v_{(i_1-1)d^{l-1}+(i_2-1)d^{l-2}+\cdots+(i_{l-1}-1)d+i_l}$. Similarly, for a $d^l \times d^l$ matrix $M$, we use $M_{i_1, i_2, \cdots, i_l, j_1, j_2, \cdots, j_l}$ to denote $\mathrm{Tensor}(M)_{i_1, i_2, \cdots, i_l, j_1, j_2, \cdots, j_l}$, or in other words, $M_{(i_1-1)d^{l-1}+(i_2-1)d^{l-2}+\cdots+(i_{l-1}-1)d+i_l,(j_1-1)d^{l-1}+(j_2-1)d^{l-2}+\cdots+(j_{l-1}-1)d+j_l}$.

Assume that $v = u^{\otimes l}$, then $v \in \mathbb{R}^{d^{\otimes l}}$, and $v$ is a symmetric tensor. Note that
$$\forall i_1, \cdots, i_l \in [d], v_{i_1, \cdots, i_l} = u_{i_1} \cdots u_{i_l}.$$
Define $M \triangleq \mathrm{vec}(v)\mathrm{vec}(v)^\top = \mathrm{vec}(u^{\otimes l})\mathrm{vec}(u^{\otimes l})^\top$, then
$$M_{i_1, \cdots, i_l, j_1, \cdots, j_l} = u_{i_1} \cdots u_{i_l} \cdot u_{j_1} \cdots u_{j_l}.$$
By Wick's theorem, we know that
$$\mathbb{E}[M_{i_1, \cdots, i_l, j_1, \cdots, j_l}] = \sum_{\sigma \in P} \prod_{t \in [l]} \mathbb{E}[u_{\sigma(2t-1)} u_{\sigma(2t)}],$$

where $P$ is the set that containing all distinct partitions of $S = \{i_1, \cdots, i_l, j_1, \cdots, j_l\}$ into $l$ pairs. Each two variables in $S$ are considered different even if the values of them are the same, e.g., $\{(i_1, i_2), (j_1, j_2)\}$ and $\{(i_1, j_2), (j_1, i_2)\}$ are different partitions even if $i_2 = j_2$. In other words, the partition is independent of the value of those variables. Thus, we can decompose matrix $M$ into the sum of $(2l-1)!!$ matrices, i.e.,
$$M = \sum_{\sigma \in P} M_\sigma.$$
Assume $\sigma_1$ is the partition $\{(i_1, j_1), \cdots, (i_l, j_l)\}$, so
$$\mathbb{E}[M_{\sigma_1}] = \prod_{t \in [l]} \mathbb{E}[u_{i_t} u_{j_t}].$$

Since $\mathbb{E}[u_{i_t} u_{j_t}] = \mathbb{I}\{i_t = j_t\}$ ($\mathbb{I}$ is the indicator function), we know that all elements on the diagonal of $\mathbb{E}[M_{\sigma_1}]$ are 1, and all other elements are 0, which means that $\mathbb{E}[M_{\sigma_1}]$ is the identity matrix. Hence,
$$b^\top \mathbb{E}[M_{\sigma_1}]b = 1.$$

Note that $b$ is a symmetric vector, meaning $b^\top \mathbb{E}[M_{\sigma_1}]b$ doesn't change if we permute $\{i_1, \cdots, i_l\}$. Thus, as long as each $i$ is paired with a $j$ in $\sigma$, we will have $b^\top \mathbb{E}[M_{\sigma_1}]b = b^\top \mathbb{E}[M_\sigma]b$. There are $n!$ such partitions, so summing them up gives us $l!$..

For any other partition $\sigma$, we can always permute $\{i_1, \cdots, i_l\}$ and $\{j_1, \cdots, j_l\}$ such that the partition becomes $\{(i_1, i_2), \cdots, (i_{2t-1}, i_{2t}), (j_1, j_2), \cdots, (j_{2t-1}, j_{2t}), (i_{2t+1}, j_{2t+1}), \cdots, (i_l, j_l)\}$. Then
$$\mathbb{E}[M_\sigma] = I_{d^{l-2t}} \otimes (ww^\top),$$
where $w \in \mathbb{R}^{d^{2t}}$ and $w_{i_1, \cdots, i_{2t}} = \mathbb{I}\{i_1 = i_2\} \cdots \mathbb{I}\{i_{2t-1} = i_{2t}\}$. Therefore, $\mathbb{E}[M_\sigma]$ is a positive semi-definite matrix, i.e., $b^\top \mathbb{E}[M_\sigma]b \geq 0$.

In a word, we can divide $\mathbb{E}[M]$ into the sum of two sets of matrices. In the symmetric sense, the first set of matrices are equivalent to identity matrices while the second set of matrices are equivalent to some semi-definite matrices. Therefore,
$$b^\top \mathbb{E}[\mathrm{vec}(u^{\otimes l})\mathrm{vec}(u^{\otimes l})^\top]b \geq l!.$$
$\square$

**Proof of Lemma 10.** Let $u \in \mathbb{R}^d$ be a standard normal vector, i.e., $u \sim \mathcal{N}(0, I_d)$, then from Theorem 2 in Vignat and Bhatnagar (2008) we know that
$$b^\top \mathbb{E}\left[\mathrm{vec}\left((u/\|u\|)^{\otimes l}\right)\mathrm{vec}\left((u/\|u\|)^{\otimes l}\right)^\top\right]b = \frac{\Gamma\left(\frac{d}{2}\right)}{2^l \Gamma\left(l + \frac{d}{2}\right)} b^\top \mathbb{E}[\mathrm{vec}(u^{\otimes l})\mathrm{vec}(u^{\otimes l})^\top]b.$$
Furthermore, from Lemma 11 we know that
$$^\top \mathbb{E}[\mathrm{vec}(u^{\otimes l})\mathrm{vec}(u^{\otimes l})^\top]b \geq l!.$$

Note that $u/\|u\|$ is distributed as a uniform vector from the unit sphere $\mathbb{S}^{d-1}$. Combining the above equality and inequality, we finish the proof of this lemma. $\square$

# B Construction of Bad Local Minimum

In this section, we construct a bad local min for the vanilla loss function with vanilla parameterization of $T$. That is, $T := \sum_{i=1}^{m} c_i u_i^{\otimes l}$ and $f_v(U,C) = 1/2 \left\| T - T^* \right\|_F^2$. Recall Theorem 2 as follows.

**Theorem 2.** *Let $f_v(U,C)$ be as defined in Equation 2. Assume $l \geq 3, d > r \geq 1$ and $m \geq r(l+1) + 1$. There exists a symmetric ground truth tensor $T^*$ with rank at most $r(l+1) + 1$ such that a local minimum with function value $l(l-1)r/4$ exists while the global minimum has function value zero.*

In our example, the model fits one direction in $T^*$ but misses all the other directions. Moving any component towards one of the missing directions would actually make the approximation worse because of the cross terms. The proof of Theorem 2 is in Section B.1.

We also extend the local min to the vanilla loss function with 2-homogeneous parameterization of $T$. That is, $T := \sum_{i=1}^{m} a_i c_i^{l-2} u_i^{\otimes l}$ and $f(U,C) = 1/2 \left\| T - T^* \right\|_F^2$. For simplicity, we assume half of the $a_i$'s are 1's.

**Theorem 4.** *Let $f(U,C) := 1/2 \left\| \sum_{i=1}^{m} a_i c_i^{l-2} u_i^{\otimes l} - T^* \right\|_F^2$. Assume $\lfloor m/2 \rfloor$ of $a_i$'s are 1's and the remaining are $-1$'s. Assume $l \geq 3, d - 2 \geq r \geq 1$ and $m \geq 4r(l+1) + 2$. There exists a symmetric ground truth tensor $T^*$ with rank at most $2r(l+1) + 2$ such that a local minimum with function value $l(l-1)r/2$ exists while the global minimum has function value zero.*

In the above Theorem, we treat $c_i$'s as separate variables from $u_i$'s. That is, at a local min $(U,C)$, we allow arbitrary perturbations to $c_i$'s regardless of the perturbations to $u_i$'s and show none of these perturbations can decrease the function value. Note our result trivially extends to the case when $c_i = 1/ \left\| u_i \right\|$ since the coupling between $c_i$ and $u_i$ only restricts the set of possible perturbations to $(U,C)$. The detailed proof of Theorem 4 is in Section B.1.

## B.1 Detailed Proofs

**Proof of Theorem 2.** In this proof, we first construct a ground truth tensor and a local min with non-zero loss. To further prove this local min is indeed spurious, we show there exists a global min with zero loss under the same ground truth tensor.

We first define the local min. For every $i \in [m]$, let $c_i$ be 1 and $u_i$ be $e_1/m^{\frac{1}{l}}$. Then, we know at this point $T = e_1^{\otimes l}$.

We define the ground truth tensor $T^*$ by defining the residual $R := T - T^*$. The residual $R$ is defined as the summation of $\hat{R}$ and all its permutation. We define $\hat{R}$ as follows,

$$\hat{R} := \sum_{j=2}^{r+1} e_j^{\otimes 2} \otimes e_1^{\otimes l-2}.$$

Then, $R$ is defined as the summation of all $\binom{l}{2}$ permutations of $\hat{R}$. It's clear that $R$ is symmetric and therefore $T^*$ is symmetric.

Let $U$ be a $d \times m$ matrix whose $i$-th row is $u_i$, and $C$ be an $m \times m$ diagonal matrix with $C_{ii} = c_i, \forall i \in [m]$. Suppose we perform a local change to $U$ and $C$ such that $U' = U + \Delta U, C' = C + \Delta C$ and $\left\| \Delta U \right\|_F, \left\| \Delta C \right\|_F \to 0$. We prove that for any $\Delta U, \Delta C$, we have $f(U',C') \geq f(U,C)$. Let's first show that the gradient at $U$ is zero, which means there is no locally first-order change on the function value.

**First-order Change** Let's first show the gradient of $f_v$ w.r.t. $U$ and $C$ at $(U,C)$ is zero. Here we first compute the gradient in terms of one column $u_i$,

$$\forall i \in [m], \nabla_{u_i} f_v(U,C) = lR(u_i^{\otimes l-1}, I)c_i = \frac{l}{m^{\frac{l-1}{l}}} R(e_1^{\otimes l-1}, I).$$

$$\forall i \in [m], \nabla_{c_i} f_v(U,C) = R(u_i^{\otimes l}) = \frac{1}{m} R(e_1^{\otimes l}).$$

In order to compute $R(e_1^{\otimes l-1}, I)$, we first consider $\hat{R}(e_1^{\otimes l-1}, I)$. We have

$$\hat{R}(e_1^{\otimes l-1}, I) = \sum_{j=2}^{r+1} e_j \langle e_j, e_1 \rangle \langle e_1, e_1 \rangle^{l-2} = 0.$$

Similarly,

$$\hat{R}(e_1^{\otimes l}) = \sum_{j=2}^{r+1} \langle e_j, e_1 \rangle^2 \langle e_1, e_1 \rangle^{l-2} = 0.$$

The computation for other permutations of $\hat{R}$ is the same. Overall, we have $\nabla f_v(U, C) = 0$.

**Second-order change** The second order change of $f_v(U, C)$ is as follows,

$$\frac{1}{2} \left\| \sum_{i=1}^m \left( c_i (\Delta u_i) \otimes u_i^{\otimes l-1} + c_i u_i \otimes (\Delta u_i) \otimes u_i^{\otimes l-2} + \cdots + c_i u_i^{\otimes l-1} \otimes (\Delta u_i) \right) + (\Delta c_i) u_i^{\otimes l} \right\|_F^2$$
$$+ \sum_{i=1}^m \left( l(l-1) R(u_i^{\otimes l-2}, \Delta u_i, \Delta u_i) c_i + l R(u_i^{\otimes l-1}, \Delta u_i) \Delta c_i \right).$$

The first term is always non-negative, and the second term can be further computed as follows:

$$l(l-1) \sum_{i=1}^m R(u_i^{l-2}, \Delta u_i, \Delta u_i) = l(l-1) \frac{1}{m^{(l-2)/l}} \sum_{i=1}^m R(e_1^{\otimes l-2}, \Delta u_i, \Delta u_i)$$
$$= l(l-1) \frac{1}{m^{(l-2)/l}} \sum_{i=1}^m \sum_{j=2}^{r+1} [\Delta u_i]_j^2.$$

Similar to the computations in the first-order change part, we know $R(u_i^{\otimes l-1}, \Delta u_i) = 0$. Therefore, the second-order change of $f(U, C)$ can be lower bounded by $l(l-1) \frac{1}{m^{(l-2)/l}} \sum_{i=1}^m \sum_{j=2}^{r+1} [\Delta u_i]_j^2$.

For any $\Delta U, \Delta C$, if there exists $i \in [m], 2 \le j \le r+1$ such that $[\Delta u_i]_j \ne 0$, we know the second order change is positive. Combining with the fact that the gradient is zero at $(U, C)$, this implies the function value increases.

Otherwise, if $[\Delta u_i]_j = 0$ for all $i \in [m]$ and all $2 \le j \le r+1$, we know $\Delta u_i \in B$ for all $i \in [m]$ where $B$ is the span of $\{e_1, e_k | r+2 \le k \le d\}$. Let the perturbed tensor be $T'$, we know $T' - T$ lies in the $B^{\otimes l}$ subspace. Note perturbing $c_i$ introduces changes in $e_1^{\otimes l}$ direction that is also in the $B^{\otimes l}$ subspace. This type of perturbation cannot decrease the function value because the residual $R$ is orthogonal with $B^{\otimes l}$ subspace.

Overall, we have proved that $(U, C)$ is a local minimizer. Notice that residual $R$ contains $r \binom{l}{2}$ orthogonal components with unit norm. Therefore, the function value at $(U, C)$ is $f(U, C) = \frac{1}{2} \|R\|_F^2 = \frac{1}{2} \times r \times \binom{l}{2} = \frac{l(l-1)r}{4}$.

**Construction of global minimizer:** Next we will show that when $m \ge r(l+1) + 1$, there exists $U$ and $C$ such that $f(U, C) = 0$. Therefore, the local minimizer we found above must be a spurious local minimizer. We only need to show that $T^*$ can be expressed as the summation of $r(l+1) + 1$ rank-one symmetric tensors.

Define $\hat{R}_j := e_j^{\otimes 2} \otimes e_1^{\otimes l-2}$, and define $R_j$ to be the sum of all $\binom{l}{2}$ permutations of $\hat{R}_j$. Then we can write $T^*$ as

$$T^* = e_1^{\otimes l} + \sum_{j=2}^{r+1} R_j.$$

Note that $R_j$ is a symmetric tensor with entries equal to 1 if the index of the entry has 2 $j$'s and $(l-2)$ 1's, and entries equal to 0 otherwise. Define $v_{i,j} := e_1 + b_{i,j} e_j$ where $b_{i,j} \in \mathbb{R}$, and consider the tensor $\bar{T}_j := \sum_{i=1}^{l+1} \bar{b}_{i,j} v_{i,j}^{\otimes l}$. Then we know that $\bar{T}_j$ is also a symmetric tensor with entries equal to $\sum_{i=1}^{l+1} \bar{b}_{i,j} b_{i,j}^k$ if the index of the entry has $k$ $j$'s and $(l-k)$ 1's, and entries equal to 0 otherwise.

Therefore, if $\forall k \in \{0, 1, \cdots, l\} \setminus \{2\}$, $\sum_{i=1}^{l+1} \bar{b}_{i,j} b_{i,j}^k = 0$ and $\sum_{i=1}^{l+1} \bar{b}_{i,j} b_{i,j}^2 = 1$, then $\bar{T}_j = R_j$. In other words, we want

$$\begin{pmatrix} 0 \\ 0 \\ 1 \\ 0 \\ \vdots \\ \vdots \\ 0 \end{pmatrix} = \begin{pmatrix} 1 & 1 & \cdots & 1 \\ b_{1,j} & b_{2,j} & \cdots & b_{l+1,j} \\ b_{1,j}^2 & b_{2,j}^2 & \cdots & b_{l+1,j}^2 \\ \vdots & \vdots & \vdots & \vdots \\ b_{1,j}^l & b_{2,j}^l & \cdots & b_{l+1,j}^l \end{pmatrix} \begin{pmatrix} \bar{b}_{1,j} \\ \bar{b}_{2,j} \\ \vdots \\ \bar{b}_{l+1,j} \end{pmatrix}.$$

Denote the matrix in the middle by $M_j$, then

$$|M_j| = \prod_{1 \leq s < t \leq l+1} (b_{s,j} - b_{t,j}).$$

Thus, as long as the $b_{i,j}$'s are mutually different, the matrix $M_j$ will be full rank, and there must exist a set of $\bar{b}_{i,j}$'s such that the equation above holds. In other words, we have shown that there exists such $b_{i,j}$'s and $\bar{b}_{i,j}$'s that $\forall 2 \leq j \leq r + 1$, $\bar{T}_j = R_j$. Therefore, we know each $R_j$ can be expressed as the summation of $(l + 1)$ rank-one symmetric tensors.

To summarize, when $m \geq r(l + 1) + 1$, we can construct $T^*$ such that there exists a local minimum with function value $\frac{l(l-1)r}{4}$ while the global minimum has function value zero.

$\square$

**Proof of Theorem 4.** The proof is very similar as the proof of Theorem 2. The only difference is that in the 2-homogeneous, all the $c_i$'s are positive and we need to rely on positive and negative $a_i$'s to fit the ground truth tensors. We need to define a slightly different ground truth tensor and bad local min.

Same as in the proof of Theorem 2, we first construct a ground truth tensor and a local min with non-zero loss. To further prove this local min is indeed spurious, we show there exists a global min with zero loss under the same ground truth tensor.

We first define the local min. Let $m' = \lfloor m/2 \rfloor$. Without loss of generality, assume $a_i = 1$ for all $i \in [m']$ and $a_i = -1$ for all $i \in [m' + 1, m]$. For any $i \in [m']$, let $u_i = \sqrt{1/m'} e_1$ and $c_i = 1/\|u_i\|$. For any $i \in [m' + 1, m]$, let $u_i = \sqrt{1/(m - m')} e_d$ and $c_i = 1/\|u_i\|$. With this choice of parameters, it's not hard to verify that $T = e_1^{\otimes l} - e_d^{\otimes l}$.

We define the ground truth tensor $T^*$ by defining the residual $R := T - T^*$. The residual $R$ is defined as the summation of $\hat{R}$ and all its permutation, where $\hat{R}$ is defined as:

$$\hat{R} := \sum_{j=2}^{r+1} \left( e_j^{\otimes 2} \otimes e_1^{\otimes l-2} - e_j^{\otimes 2} \otimes e_d^{\otimes l-2} \right).$$

Since we assume $r \leq d - 2$, we know $r + 1 \leq d - 1$ and $e_j$ is orthogonal with $e_1, e_d$ for all $2 \leq j \leq r + 1$. Then, $R$ is defined as the summation of all $\binom{l}{2}$ permutations of $\hat{R}$. It's clear that $R$ is symmetric and $T^*$ is also symmetric.

Suppose we perform a local change to $U$ and $C$ such that $U' = U + \Delta U, C' = C + \Delta C$ and $\|\Delta U\|_F, \|\Delta C\|_F \to 0$, where $\Delta C$ is a diagonal matrix. We prove that for all possible $\Delta U, \Delta C$, $f(U', C') \geq f(U, C)$.

**First-order Change** Let's first show the derivative of $f$ in terms of all $u_i$'s and $c_i$'s at $(U, C)$ is zero. This means there is no first order decrease direction at $(U, C)$.

For any $i \in [m']$, we can compute the derivative in terms of $u_i$ and $c_i$:

$$\nabla_{u_i} f(U, C) = lR(u_i^{\otimes l-1}, I) c_i^{l-2} = \frac{l}{\sqrt{m'}} R(e_1^{\otimes l-1}, I),$$

$$\nabla_{c_i} f(U, C) = (l - 2) R(u_i^{\otimes l}) c_i^{l-3} = \frac{l - 2}{(m')^{3/2}} R(e_1^{\otimes l}).$$

It's not hard to verify that $R(e_1^{\otimes l-1}, I) = 0$ and $R(e_1^{\otimes l}) = 0$ using the orthogonality between $e_1$ and $e_j$ for all $2 \leq j \leq r + 1$. For the same reason, we also have $\nabla_{u_i} f(U, C) = 0, \nabla_{c_i} f(U, C) = 0$ for all $i \in [m' + 1, 2m]$.

**Second-order change**   The second order change of $f(U', C')$ compared with $f(U, C)$ is as follows,

$$\frac{1}{2} \left\| \sum_{i=1}^{m} a_i \left( c_i^{l-2} \left( (\Delta u_i) \otimes u_i^{\otimes l-1} + u_i \otimes (\Delta u_i) \otimes u_i^{\otimes l-2} + \cdots + u_i^{\otimes l-1} \otimes (\Delta u_i) \right) + (l-2) c_i^{l-3} (\Delta c_i) u_i^{\otimes l} \right) \right\|_F^2$$

$$+ \sum_{i=1}^{m} a_i \left( l(l-1) R(u_i^{\otimes l-2}, \Delta u_i, \Delta u_i) c_i^{l-2} + l(l-2) R(u_i^{\otimes l-1}, \Delta u_i) c_i^{l-3} \Delta c_i + (l-2)(l-3) R(u_i^{\otimes l}) c_i^{l-4} (\Delta c_i)^2 \right).$$

(When $l = 3$, we do not have the $c_i^{l-4}$ term)

The first term is always non-negative. By the previous argument, we also know $R(u_i^{\otimes l-1}, \Delta u_i) = 0$ and $R(u_i^{\otimes l}) = 0$. Therefore, the second order change is lower bounded by $\sum_{i=1}^{m} a_i l(l-1) R(u_i^{\otimes l-2}, \Delta u_i, \Delta u_i) c_i^{l-2}$. Let's first consider the components from 1 to $m'$ for which $a_i = 1$,

$$l(l-1) a_i \sum_{i=1}^{m'} R(u_i^{\otimes l-2}, \Delta u_i, \Delta u_i) c_i^{l-2} = l(l-1) \sum_{i=1}^{m'} R(e_1^{\otimes l-2}, \Delta u_i, \Delta u_i)$$

$$= l(l-1) \sum_{i=1}^{m'} \sum_{j=2}^{r+1} [\Delta u_i]_j^2.$$

For the components from $m' + 1$ to $m$, we have

$$l(l-1) a_i \sum_{i=m'+1}^{m} R(u_i^{\otimes l-2}, \Delta u_i, \Delta u_i) c_i^{l-2} = l(l-1) \sum_{i=m'+1}^{m} -R(e_d^{\otimes l-2}, \Delta u_i, \Delta u_i)$$

$$= l(l-1) \sum_{i=m'+1}^{m} \sum_{j=2}^{r+1} [\Delta u_i]_j^2.$$

Overall, we have

$$\sum_{i=1}^{m} a_i l(l-1) R(u_i^{\otimes l-2}, \Delta u_i, \Delta u_i) c_i^{l-2} \geq l(l-1) \sum_{i=1}^{m} \sum_{j=2}^{r+1} [\Delta u_i]_j^2.$$

For any $\Delta U, \Delta C$, if there exists $i \in [m], 2 \leq j \leq r+1$ such that $[\Delta u_i]_j \neq 0$, we know the second order change is positive. Combining with the fact that the gradient is zero at $(U, C)$, this implies the function value increases.

Otherwise, if $[\Delta u_i]_j = 0$ for all $i \in [m]$ and all $2 \leq j \leq r+1$, we know $\Delta u_i \in B$ for all $i \in [m]$ where $B$ is the span of $\{e_1, e_k | r+2 \leq k \leq d\}$. Let the perturbed tensor be $T'$, we know $T' - T$ lies in the $B^{\otimes l}$ subspace. Note perturbing $c_i$ introduces changes in $e_1^{\otimes l}$ direction or $e_d^{\otimes l}$ direction that are also in the $B^{\otimes l}$ subspace. This type of perturbation cannot decrease the function value because the residual $R$ is orthogonal with $B^{\otimes l}$ subspace.

Overall, we have proved that $(U, C)$ is a local minimizer. Notice that residual $R$ contains $2r \binom{l}{2}$ orthogonal components with unit norm. Therefore, the function value at $(U, C)$ is $f(U, C) = \frac{1}{2} \|R\|_F^2 = \frac{1}{2} \times 2r \times \binom{l}{2} = \frac{l(l-1)r}{2}$.

**Construction of global minimizer:**   Next, we show as long as $m \geq 4r(l+1) + 2$, there exists parameters $(U, C)$ such that $f(U, C) = 0$. To prove this, we first write $T^*$ as summation of rank-one symmetric tensors.

For any $2 \leq j \leq r+1$, define $\hat{R}_{j,1} := e_j^{\otimes 2} \otimes e_1^{\otimes l-2}$ and $\hat{R}_{j,d} = e_j^{\otimes 2} \otimes e_d^{\otimes l-2}$, and define $R_{j,1}, R_{j,d}$ to be the sum of all $\binom{l}{2}$ permutations of $\hat{R}_{j,1}$ and $\hat{R}_{j,d}$ respectively. Then we can write $T^*$ as

$$T^* = e_1^{\otimes l} - e_d^{\otimes l} - \sum_{j=2}^{r+1} R_{j,1} + \sum_{j=2}^{r+1} R_{j,d}.$$

Same as in the proof of Theorem 2, we can show each $R_{j,1}$ or $R_{j,d}$ can be written as the sum of $(l+1)$ rank-one symmetric tensors.

Therefore, we know the ground truth $T^*$ can be expressed as the summation of $2 + 2r(l+1)$ rank-one symmetric tensors. For each component, we can re-scale it to make it fit the form of $a_i c_i^{l-2} u_i^{\otimes l}$, with $a_i = \pm 1$ and $c_i = 1/\|u_i\|$. In these rank-one tensors, at most $2r(l+1) + 1$ has positive (negative) $a_i$. So, as long as $m' \geq 2r(l+1) + 1$ or $m \geq 4r(l+1) + 2$ our model is able to fit this ground truth tensor and achieve zero loss.

To summarize, when $m \geq 4r(l+1) + 2$, we can construct $T^*$ with rank at most $2r(l+1) + 2$ such that there exists a local minimum with function value $\frac{l(l-1)r}{2}$ while the global minimum has function value zero. $\qquad\square$

## C  Detailed Proofs of Theorem 3

In this section, we give the proof of Theorem 3. We first state a formal version of Theorem 3.

**Theorem 5.** *Given any target accuracy $\epsilon > 0$, there exists $m = O\left(\frac{r^{2.5l}}{\epsilon^5}\log(d/\epsilon)\right)$, $\lambda = O\left(\frac{\epsilon}{r^{0.5l}}\right)$, $\delta = O\left(\frac{\epsilon^{5l-1.5}}{d^{l-1.5}(\log(d/\epsilon))^{l+0.5}r^{2.5l^2-0.75l}}\right)$, $\eta = O\left(\frac{\epsilon^{15l-4.5}}{d^{3l-4.5}(\log(d/\epsilon))^{3l+1.5}r^{7.5l^2-2.25l}}\right)$ and $H = O\left(\frac{d^{3l-4.5}(\log(d/\epsilon))^{3l+2.5}r^{7.5l^2-1.75l}}{\epsilon^{15l-3.5}}\right)$ such that with probability at least $0.99$, our algorithm finds a tensor $T$ satisfying*

$$\|T - T^*\|_F \leq \epsilon,$$

*within $K = O\left(\frac{r^{2l}}{\epsilon^4}\log(d/\epsilon)\right)$ epochs.*

We follow the proof strategy outlined in Section 5.

As we discussed in Challenge 2, bad local minima exist for our loss function. Therefore, gradient descent might get stuck at a bad local minima. This issue is fixed in our algorithm by re-initializing one component at the beginning of each epoch. In Lemma 1, we show as long as the objective is large, there is at least a constant probability to improve the objective within one epoch. We state the formal version of Lemma 1 as follows. The proof of Lemma 12 is in Section C.2.

**Lemma 12.** *Let $(U'_0, \bar{C}'_0)$ and $(U_H, \bar{C}_H)$ be the parameters at the beginning of an epoch and the parameters at the end of the same epoch. For the target accuracy $\epsilon > 0$ in Theorem 5, assume $K \leq \frac{\lambda m}{14}$ and $\|T'_0 - T^*\|_F \geq \epsilon$ where $T'_0$ is the tensor with parameters $(U'_0, \bar{C}'_0)$. There exists $m = O\left(\frac{r^{2.5l}}{\epsilon^5}\log(d/\epsilon)\right)$, $\lambda = O\left(\frac{\epsilon}{r^{0.5l}}\right)$, $\delta = O\left(\frac{\epsilon^{5l-1.5}}{d^{l-1.5}(\log(d/\epsilon))^{l+0.5}r^{2.5l^2-0.75l}}\right)$, $\eta = O\left(\frac{\epsilon^{15l-4.5}}{d^{3l-4.5}(\log(d/\epsilon))^{3l+1.5}r^{7.5l^2-2.25l}}\right)$, $H = O\left(\frac{d^{3l-4.5}(\log(d/\epsilon))^{3l+2.5}r^{7.5l^2-1.75l}}{\epsilon^{15l-3.5}}\right)$, such that with probability at least $\frac{1}{6}$, we have*

$$f(U_H, \bar{C}_H) - f(U'_0, \bar{C}'_0) \leq -\Omega\left(\frac{\epsilon^4}{r^{2l}\log(d/\epsilon)}\right).$$

We compliment this lemma by showing that even if an epoch does not improve the objective, it will not increase the function value by too much. The formal version of Lemma 2 is as follows. We prove Lemma 13 in Section C.1.

**Lemma 13.** *Assume $K \leq \frac{\lambda m}{14}$, $\delta \leq \frac{\mu_1 \epsilon}{m^{\frac{3}{4}} d^{\frac{l-2}{2}}(m+K)^{\frac{l-1}{2}}\lambda^{\frac{1}{2}}}$, and $\eta \leq \frac{\mu_2 \lambda}{m^{\frac{1}{2}} d^{\frac{l-1}{2}}(m+K)^{\frac{l-2}{2}}}$ for some constants $\mu_1, \mu_2$, and $\frac{10}{m} \leq \lambda \leq 1$. Let $(U'_0, \bar{C}'_0)$ and $(U_H, \bar{C}_H)$ be the parameters at the beginning of an epoch and the parameters at the end of the same epoch. Assume $f(U'_0, \bar{C}'_0) \geq \epsilon^2$, where $\epsilon$ is the target accuracy in Theorem 5. Then we have*

$$f(U_H, \bar{C}_H) - f(U'_0, \bar{C}'_0) \leq O\left(\frac{1}{\lambda m}\right).$$

From these two lemmas, we know that in each epoch, the loss function can decrease by $\Omega\left(\frac{\epsilon^4}{r^{2l}\log(d/\epsilon)}\right)$ with probability at least $\frac{1}{6}$, and even if we fail to decrease the function value,

the increase of function value is at most $O\left(\frac{1}{\lambda m}\right)$. Therefore, choosing a large enough $m$, the function value decrease will dominate the increase. This allows us to prove Theorem 5.

**Proof of Theorem 5.** We use a contradiction proof to show that with high probability our algorithm finds a tensor $T$ satisfying $\|T - T^*\|_F \leq \epsilon$ within $K$ epochs. For the sake of contradiction, we assume $\|T - T^*\|_F > \epsilon$ through the first $K$ epochs. Under this assumption, we show with high probability the function value will decrease below zero.

Note that under the choice of parameters of this theorem, all the conditions of Lemma 12 and Lemma 13 are satisfied. By Lemma 12, we know that with probability at least $1/6$, the function value decreases by at least $\Lambda := \Omega\left(\frac{\epsilon^4}{r^{2l} \log(d/\epsilon)}\right)$ in each epoch. By Lemma 13, we show that the function value at most increases by $\Lambda' := O(\frac{1}{\lambda m})$ in each epoch. Using our choice of the parameters in Theorem 5, we know that $O(\frac{1}{\lambda m}) = O\left(\frac{\epsilon^4}{r^{2l} \log(d/\epsilon)}\right)$. Choosing a large enough constant factor for $m$ ensures that $\Lambda' \leq \frac{\Lambda}{10}$.

For each $1 \leq k \leq K$, let $\mathcal{E}_k$ be the event that in the beginning of the $k$-th epoch, the reinitialized component $ac_0^{l-2}u_0^l$ has good correlation with the residual (see Lemma 18) and $\|P_S u_0\| \geq \frac{\mu\delta}{\sqrt{d}}$, where $\mu$ is some constant. We know $\mathcal{E}_k$'s are independent with each other and $\Pr[\mathcal{E}_k] \geq 1/6$. By Hoeffding's inequality, we know as long as $K \geq \mu'$ for certain constant $\mu'$, we have $\sum_{k=1}^{K} \mathbb{1}_{\mathcal{E}_k} \geq K/7$ with probability at least $0.99$, where $\mathbb{1}_{\mathcal{E}_k}$ is the indicator function of event $\mathcal{E}_k$.

By the proof of Lemma 12, we know conditioning on $\mathcal{E}_k$, the function value decreases by at least $\Lambda$ in the $k$-th epoch. Since $\sum_{k=1}^{K} \mathbb{1}_{\mathcal{E}_k} \geq K/7$, we know the total function value decrease is at least $K\Lambda/7 - K\Lambda/10 = K\Omega\left(\frac{\epsilon^4}{r^{2l} \log(d/\epsilon)}\right)$. Therefore, there exists $K = O\left(\frac{r^{2l} \log(d/\epsilon)}{\epsilon^4}\right)$ such that $K\Lambda/7 - K\Lambda/10 \geq 4$.

By the analysis in Lemma 16, the function value is upper bounded by 3 at initialization. However, with probability at least $0.99$, the decrease of the function value is at least 4, meaning that the function value must be negative, which is a contradiction. Therefore, we know that with probability at least $0.99$, our algorithm finds a tensor $T$ satisfying $\|T - T^*\|_F \leq \epsilon$ within $K = O\left(\frac{r^{2l} \log(d/\epsilon)}{\epsilon^4}\right)$ epochs. $\qquad\square$

## C.1 Upper bound on function increase

In this section, we prove Lemma 13.

To prove the increase of $f$ is bounded in one epoch, we identify all the possible ways that the loss can increase and upper bound each of them. We first show that a normal step (without re-initialization or scalar mode switch) of the algorithm will not increase the objective function. Note that many parts of our proofs rely on an upperbound on function value. To get such a bound the proof includes an induction component: when we prove Lemma 14 and Lemma 15, we assume that the function value is upper bounded by a constant, and we will inductively prove that these conditions are satisfied in Lemma 16. This induction ensures that the conclusions of all the lemmas in this section hold throughout the entire algorithm.

The following lemma is a formal version of Lemma 3 in the main text.

**Lemma 14.** *Let $(U, \bar{C})$ be the parameters at the beginning of one iteration and let $U', \bar{C}'$ be the updated parameters (before potential scalar mode switch). Assuming $f(U, \bar{C}) \leq 10, \lambda \leq 1$, there exists constants $\mu_1, \mu_2$ such that*

$$f(U', \bar{C}') - f(U, \bar{C}) \leq -\frac{\eta}{l} \left\|\nabla_U f(U, \bar{C})\right\|_F^2$$

*as long as $\delta \leq \frac{\mu_1}{m^{\frac{1}{4}} \sqrt{\lambda} d^{\frac{l-2}{2}} (m+K)^{\frac{l-1}{2}}}, \eta \leq \frac{\mu_2 \lambda}{m^{\frac{1}{2}} d^{\frac{l-1}{2}} (m+K)^{\frac{l-2}{2}}}.$*

Recall that in an iteration, we first update $U$ by gradient descent, then update $C$ and $\hat{C}$ by the updated value of $U$. The gradient descent step on $U$ cannot increase the function value as long as the step size is small enough. The update on $C$ and $\hat{C}$ can potentially increase the function value. In the

proof of Lemma 14, we show the increase due to updating $C$ and $\hat{C}$ is proportional to the decrease by updating $U$ and smaller in scale.

**Proof of Lemma 14.**

According to the algorithm, each iteration contains two steps: update $U$ as $U' \leftarrow U - \eta \nabla_U f(U, \bar{C})$; update $c_i$ and $\hat{c}_i$ as $c_i' = c_i \frac{\|u_i\|}{\|u_i'\|}$ and $\hat{c}_i' = \hat{c}_i \frac{\|u_i\|}{\|u_i'\|}$. We can divide the function value change into these two steps: $f(U', \bar{C}') - f(U, \bar{C}) = (f(U', \bar{C}) - f(U, \bar{C})) + (f(U', \bar{C}') - f(U', \bar{C}))$. We will show that the function value decrease in the first step and does not increase by too much in the second step. At the end, we will combine them to show that overall the function value decreases.

Since we assume $f(U, \bar{C}) \leq 10$. According to the definition of the loss function, we know $\|T - T^*\|_F \leq \sqrt{20}, \sum_{i=1}^m \|u_i\|^2 \leq \frac{10}{\lambda}$. We also know that $\sum_{i=1}^m \|u_i\|^4 \leq \left( \sum_{i=1}^m \|u_i\|^2 \right)^2 \leq \frac{100}{\lambda^2}$. For convenience, denote $\Gamma = 10$, $M_4^2 := \frac{100}{\lambda^2}$ and $M_2^2 := \frac{10}{\lambda}$.

$f(U', \bar{C}) - f(U, \bar{C})$ **is negative:** In the first step, we update $U$ by gradient descent, which should decrease the function value as long as we choose the step size to be small enough. To prove that an inverse polynomially step size suffices, we need to bound the second derivative of $f$ in terms of $U$ at $(U'', \bar{C})$ for any $U'' \in \{(1-\theta)U + \theta U' | 0 \leq \theta \leq 1\}$. Let $\mathcal{H}''$ be the Hessian of $f$ in terms of $U$ at $(U'', \bar{C})$. We will bound the Frobenius norm of $\mathcal{H}''$.

Let's first show that $\|u_i''\| \leq (1 + 1/(4l)) \|u_i\|$ when $\eta$ is small enough. Recall the derivative in $u_i$ is,

$$\nabla_{u_i} f(U, \bar{C}) = l(T - T^*)(u_i^{\otimes(l-1)}, I)c_i^{l-2}a_i + \lambda l u_i.$$

Therefore, we can bound the derivative as

$$\left\| \nabla_{u_i} f(U, \bar{C}) \right\| \leq \left( l\sqrt{2\Gamma}(\sqrt{d(m+K)})^{l-2} + \lambda l \right) \|u_i\|.$$

Thus, as long as $\eta \leq \frac{1}{4l^2(\sqrt{2\Gamma}(\sqrt{d(m+K)})^{l-2}+\lambda)}$, we have

$$\eta \left\| \nabla_{u_i} f(U, \bar{C}) \right\| \leq \eta \left( l\sqrt{2\Gamma}(\sqrt{d(m+K)})^{l-2} + \lambda l \right) \|u_i\| \leq \frac{1}{4l} \|u_i\|.$$

Since $u_i'' = u_i - \theta \eta \nabla_{u_i} f(U, \bar{C})$ for $0 \leq \theta \leq 1$, we know that

$$\|u_i''\| \leq \|u_i\| + \eta \left\| \nabla_{u_i} f(U, \bar{C}) \right\|$$

$$\leq (1 + \frac{1}{4l}) \|u_i\|,$$

Let $T''$ be the tensor parameterized by $(U'', \bar{C})$. We can bound $\|T'' - T^*\|_F$ as follows,

$$\|T'' - T^*\|_F \leq \|T - T^*\|_F + \sum_{i=1}^m \sum_{k=1}^l \binom{l}{k} \|u_i\|^{l-k} \left\| \eta \nabla_{u_i} f(U, \bar{C}) \right\|^k c_i^{l-2}$$

$$\leq \|T - T^*\|_F + \sum_{i=1}^m \sum_{k=1}^l \binom{l}{k} \frac{1}{l^k} \|u_i\|^l c_i^{l-2}$$

$$\leq \|T - T^*\|_F + l \sum_{i=1}^m \left( 4(\sqrt{d(m+K)})^{l-2}(m+K)\delta^2 + \|u_i\|^2 \right)$$

$$\leq \|T - T^*\|_F + 4lm(\sqrt{d(m+K)})^{l-2}(m+K)\delta^2 + lM_2^2$$

$$\leq \sqrt{2\Gamma} + 2lM_2^2,$$

where the last inequality assumes $\delta \leq \frac{M_2}{\sqrt{4m(\sqrt{d(m+K)})^{l-2}(m+K)}}$. For convenience, denote $\beta := \sqrt{2\Gamma} + 2lM_2^2$.

With the bound on $\|T'' - T^*\|$ and $\|u_i''\|$, we are ready to bound the Frobenius norm of $\mathcal{H}''$. For each $i \in [m]$, we have

$$\frac{\partial}{\partial u_i} f(U'', \bar{C}) = l(T'' - T^*)((u_i'')^{l-1}, I)c_i^{l-2}a_i + \lambda l \left( \frac{\|u_i''\|}{\|u_i\|} \right)^{l-2} u_i''. \tag{3}$$

We know $\mathcal{H}''$ is a $dm \times dm$ matrix that contains $m \times m$ block matrices with dimension $d \times d$. Each block corresponds to the second-order derivative of $f(U'', \bar{C})$ in terms of $u_i, u_j$. We will bound the Frobenius norm of $\mathcal{H}''$ by bounding the Frobenius norm of each block.

For each $i$, we can compute $\frac{\partial^2}{\partial u_i \partial u_i} f(U'', \bar{C})$ as follows,

$$\frac{\partial^2}{\partial u_i \partial u_i} f(U'', \bar{C}) = l(l-1)(T'' - T^*)((u_i'')^{l-2}, I, I)c_i^{l-2}a_i + l^2 c_i^{2l-4} \|u_i''\|^{2l-4} u_i'' \otimes u_i''$$
$$+ \lambda l \left( \frac{\|u_i''\|}{\|u_i\|} \right)^{l-2} I + \lambda l(l-2) \frac{\|u_i''\|^{l-4} u_i'' \otimes u_i''}{\|u_i\|^{l-2}}.$$

For the first term, we have $l(l-1) \left\| (T'' - T^*)((u_i'')^{l-2}, I, I)c_i^{l-2} \right\|_F \leq l^2 \sqrt{e}\beta(\sqrt{d(m+K)})^{l-2}$ since $\|u_i''\| / \|u_i\| \leq (1 + 1/(4l))$.

For the second term, we have

$$l^2 \left\| c_i^{2l-4} \|u_i''\|^{2l-4} u_i'' \otimes u_i'' \right\|_F \leq l^2 \sqrt{e} \max\{4(d(m+K))^{l-2}(m+K)\delta^2, \|u_i\|^2\}.$$

For the third term, we have

$$\left\| \lambda l \left( \frac{\|u_i''\|}{\|u_i\|} \right)^{l-2} \text{vec}(I) \right\|_F \leq \lambda l \sqrt{e}\sqrt{d}.$$

For the fourth term, we have

$$\left\| \lambda l(l-2) \frac{\|u_i''\|^{l-4} u_i'' \otimes u_i''}{\|u_i\|^{l-2}} \right\|_F \leq \lambda l^2 \sqrt{e}.$$

Combing the bounds on these terms and assuming $\lambda \leq 1$, we have

$$\left\| \frac{\partial^2}{\partial u_i \partial u_i} f(U'', \bar{C}) \right\|_F \leq l^2 \sqrt{e}\beta(\sqrt{d(m+K)})^{l-2} + l^2 \sqrt{e} \max\{4d^{l-2}(m+K)^{l-1}\delta^2, \|u_i\|^2\} + 2l^2 \sqrt{e}\sqrt{d}.$$

Thus,

$$\left\| \frac{\partial^2}{\partial u_i \partial u_i} f(U'', \bar{C}) \right\|_F^2 \leq 3el^4 \beta^2 d^{l-2}(m+K)^{l-2} + 3el^4 \max\{16d^{2l-4}(m+K)^{2l-2}\delta^4, \|u_i\|^4\} + 12el^4 d$$
$$\leq 15el^4 \beta^2 d^{l-2}(m+K)^{l-2} + 3el^4 \max\{16d^{2l-4}(m+K)^{2l-2}\delta^4, \|u_i\|^4\}.$$

For each pair of $i \neq j$, we can compute $\frac{\partial^2}{\partial u_i \partial u_j} f(U'', \bar{C})$ as follows

$$\frac{\partial^2}{\partial u_i \partial u_j} f(U'', C) = l^2 a_i a_j c_i^{l-2} c_j^{l-2} \langle u_i'', u_j'' \rangle^{l-2} u_i'' \otimes u_j''.$$

The Frobenius norm square can be bounded as

$$\left\| \frac{\partial^2}{\partial u_i \partial u_j} f(U'', \bar{C}) \right\|_F^2 \leq el^4 \max\{4d^{l-2}(m+K)^{l-1}\delta^2, \|u_i\|^2\} \max\{4d^{l-2}(m+K)^{l-1}\delta^2, \|u_j\|^2\}$$
$$\leq el^4 \max\{\max\{\|u_i\|^2, \|u_j\|^2\}^2, 16d^{2l-4}(m+K)^{2l-2}\delta^4\}$$
$$\leq el^4 \left( \|u_i\|^4 + \|u_j\|^4 + 16d^{2l-4}(m+K)^{2l-2}\delta^4 \right)$$

Summing over the bounds on blocks, we can bound the Frobeneius norm of $\mathcal{H}''$,

$$\|\mathcal{H}''\|_F^2 = \sum_{i,j} \left\| \frac{\partial^2}{\partial u_i \partial u_j} f(U'', \bar{C}) \right\|_F^2$$

$$\leq 15 eml^4 \beta^2 d^{l-2}(m+K)^{l-2} + 48 eml^4 d^{2l-4}(m+K)^{2l-2}\delta^4 + 3el^4 \sum_{i=1}^{m} \|u_i\|^4$$

$$+ (m-1)el^4 \sum_{i=1}^{m} \|u_i\|^4 + 16el^4 m(m-1)d^{2l-4}(m+K)^{2l-2}\delta^4$$

$$= 15 eml^4 \beta^2 d^{l-2}(m+K)^{l-2} + 16 el^4 m(m+2)d^{2l-4}(m+K)^{2l-2}\delta^4 + (m+2)el^4 \sum_{i=1}^{m} \|u_i\|^4$$

$$\leq 15 eml^4 \beta^2 d^{l-2}(m+K)^{l-2} + 2(m+2)el^4 M_4^2$$

where the last inequality assumes $\delta \leq \left( \frac{M_4^2}{16 m d^{2l-4}(m+K)^{2l-2}} \right)^{1/4}$.

Denoting $L_1 := \sqrt{15 eml^4 \beta^2 d^{l-2}(m+K)^{l-2} + 2(m+2)el^4 M_4^2}$, we have

$$f(U', \bar{C}) - f(U, \bar{C}) \leq -\eta \left\| \nabla_U f(U, \bar{C}) \right\|_F^2 + \frac{\eta^2 L_1}{2} \left\| \nabla_U f(U, \bar{C}) \right\|_F^2$$

$f(U', \bar{C}') - f(U', \bar{C})$ **is bounded:** Next, we show that setting $c_i'$ as $c_i \frac{\|u_i\|}{\|u_i'\|}$ and $\hat{c}_i'$ as $\hat{c}_i \frac{\|u_i\|}{\|u_i'\|}$ does not increase the function value by too much. We use $\nabla_{\hat{u}_i} f$ to denote the gradient of $u_i$ through $c_i$ and $\hat{c}_i$, which means

$$\nabla_{\hat{u}_i} f = \frac{\partial f}{\partial c_i} \frac{\partial c_i}{\partial u_i} + \frac{\partial f}{\partial \hat{c}_i} \frac{\partial \hat{c}_i}{\partial u_i}.$$

In the following we first bound the Frobenius norm of the Hessian of $f$ in terms of $\hat{U}$ evaluated at $(U', \bar{C}'')$ for any $C'' \in \{\text{diag}(c_1'', \cdots, c_m'') | c_i'' = c_i \frac{\|u_i\|}{\|(1-\theta)u_i + \theta u_i'\|}, 0 \leq \theta \leq 1\}$ and $\hat{C}'' \in \{\text{diag}(\hat{c}_1'', \cdots, \hat{c}_m'') | \hat{c}_i'' = \hat{c}_i \frac{\|u_i\|}{\|(1-\theta)u_i + \theta u_i'\|}, 0 \leq \theta \leq 1\}$. We denote the Hessian at $(U', \bar{C}'')$ as $\hat{\mathcal{H}}''$, which is a $md \times md$ matrix. Hessian $\hat{\mathcal{H}}''$ contains $m \times m$ blocks with dimension $d \times d$, each of which corresponds to $\frac{\partial^2}{\partial \hat{u}_i \partial \hat{u}_j} f(U', \bar{C}'')$ for some $(i, j) \in [m] \times [m]$.

Note that $\frac{\|u_i'\|}{\|u_i''\|} \leq 1 + 1/l$ since $\|u_i' - u_i\| \leq 1/(4l) \|u_i\|$. Let $T'''$ be the tensor corresponds to $(U', \bar{C}'')$, we can bound $\|T''' - T^*\|_F$ as follows,

$$\|T''' - T^*\|_F \leq \left\| \sum_{i=1}^{m} a_i (c_i'')^{l-2} (u_i')^{\otimes l} \right\|_F + \|T^*\|_F$$

$$\leq e \left( \sum_{i=1}^{m} \|u_i\|^2 + 4(\sqrt{d(m+K)})^{l-2} m(m+K)\delta^2 \right) + 1$$

$$\leq 2eM_2^2 + 1,$$

where the last step assumes $\delta \leq \frac{M_2}{\sqrt{4(\sqrt{d(m+K)})^{l-2}m(m+K)}}$. For convenience, denote $\alpha := 2eM_2^2 + 1$.

Let's first compute the derivative of $f$ in terms of $\hat{u}_i$,

$$\frac{\partial}{\partial \hat{u}_i} f(U', \bar{C}'') \tag{4}$$

$$= -(l-2)a_i(T''' - T^*)((u_i')^{\otimes l}) \frac{u_i''}{\|u_i''\|^l} \left( (\sqrt{d(m+K)})^{l-2} \mathbb{1}_{c_i'' = \sqrt{d(m+K)}/\|u_i''\|} + \mathbb{1}_{c_i'' = 1/\|u_i''\|} \right)$$

$$- \lambda(l-2) \frac{u_i''}{\|u_i''\|^l} \|u_i'\|^l. \tag{5}$$

For each $i$, we have

$$\frac{\partial^2}{\partial \hat{u}_i \partial \hat{u}_i} f(U', \bar{C}'')$$

$$= -(l-2)a_i(T''' - T^*)((u_i')^{\otimes l})\frac{I}{\|u_i''\|^l}\left((\sqrt{d(m+K)})^{l-2}\mathbb{1}_{c_i'' = \sqrt{d(m+K)}/\|u_i''\|} + \mathbb{1}_{c_i''=1/\|u_i''\|}\right)$$

$$+ l(l-2)a_i(T''' - T^*)((u_i')^{\otimes l})\frac{u_i''(u_i'')^\top}{\|u_i''\|^{l+2}}\left((\sqrt{d(m+K)})^{l-2}\mathbb{1}_{c_i'' = \sqrt{d(m+K)}/\|u_i''\|} + \mathbb{1}_{c_i''=1/\|u_i''\|}\right)$$

$$+ (l-2)^2\|u_i'\|^{2l}\frac{u_i''(u_i'')^\top}{\|u_i''\|^{2l}}\left(d^{l-2}(m+K)^{l-2}\mathbb{1}_{c_i'' = \sqrt{d(m+K)}/\|u_i''\|} + \mathbb{1}_{c_i''=1/\|u_i''\|}\right)$$

$$- \lambda(l-2)\frac{I}{\|u_i''\|^l}\|u_i'\|^l$$

$$+ \lambda l(l-2)\frac{u_i''(u_i'')^\top}{\|u_i''\|^{l+2}}\|u_i'\|^l.$$

We bound its Frobenius norm square by

$$\left\|\frac{\partial^2}{\partial \hat{u}_i \partial \hat{u}_i} f(U', \bar{C}'')\right\|_F^2$$

$$\leq 5\left(l^2\alpha^2 e^2 d^{l-1}(m+K)^{l-2} + l^4\alpha^2 e^2 d^{l-2}(m+K)^{l-2}\right)$$

$$+ 5\left(l^4 e^4 \max\{\|u_i\|^4, 16(m+K)^{2l-2}\delta^4 d^{2l-4}\} + \lambda^2 l^2 de^2 + \lambda^2 l^4 e^2\right)$$

$$\leq 20e^2\alpha^2 l^4 d^{l-1}(m+K)^{l-2} + 5e^4 l^4\left(\|u_i\|^4 + 16(m+K)^{2l-2}\delta^4 d^{2l-4}\right),$$

where we assume that $\lambda \leq 1$.

For $i \neq j$, we have

$$\frac{\partial^2}{\partial \hat{u}_i \partial \hat{u}_j} f(U', \bar{C}'')$$

$$= (l-2)^2(\langle u_i', u_j'\rangle^l)\frac{u_i''(u_j'')^\top}{\|u_i''\|^l\|u_j''\|^l}$$

$$\cdot \left((\sqrt{d(m+K)})^{l-2}\mathbb{1}_{c_i'' = \sqrt{d(m+K)}/\|u_i''\|} + \mathbb{1}_{c_i''=1/\|u_i''\|}\right)$$

$$\cdot \left((\sqrt{d(m+K)})^{l-2}\mathbb{1}_{c_j'' = \sqrt{d(m+K)}/\|u_j''\|} + \mathbb{1}_{c_j''=1/\|u_j''\|}\right)$$

We can bound its Frobenius norm by

$$\left\|\frac{\partial^2}{\partial \hat{u}_i \partial \hat{u}_j} f(U', \bar{C}'')\right\|_F^2 \leq l^4 e^4 \max\{4d^{l-2}(m+K)^{l-1}\delta^2, \|u_i\|^2\}\max\{4d^{l-2}(m+K)^{l-1}\delta^2, \|u_j\|^2\}$$

$$\leq l^4 e^4\left(\|u_i\|^4 + \|u_j\|^4 + 16(m+K)^{2l-2}\delta^4 d^{2l-4}\right)$$

Combing the bounds on all blocks, we have

$$
\begin{aligned}
\left\|\hat{\mathcal{H}}''\right\|_F^2 &\leq \sum_{i=1}^m \left(20e^2\alpha^2 l^4 d^{l-1}(m+K)^{l-2} + 5e^4 l^4 \left(\|u_i\|^4 + 16(m+K)^{2l-2}\delta^4 d^{2l-4}\right)\right) \\
&\quad + \sum_{\substack{i,j\in[m] \\ i\neq j}} l^4 e^4 \left(\|u_i\|^4 + \|u_j\|^4 + 16(m+K)^{2l-2}\delta^4 d^{2l-4}\right) \\
&\leq 20me^2\alpha^2 l^4 d^{l-1}(m+K)^{l-2} + 80ml^4 e^4(m+K)^{2l-2}\delta^4 d^{2l-4} + 5l^4 e^4 \sum_{i=1}^m \|u_i\|^4 \\
&\quad + 16m^2 l^4 e^4(m+K)^{2l-2}\delta^4 d^{2l-4} + 2l^4 e^4 m \sum_{i=1}^m \|u_i\|^4 \\
&\leq 20me^2\alpha^2 l^4 d^{l-1}(m+K)^{l-2} + 7l^4 e^4 m M_4^2 + 96m^2 l^4 e^4(m+K)^{2l-2}\delta^4 d^{2l-4} \\
&\leq 20me^2\alpha^2 l^4 d^{l-1}(m+K)^{l-2} + 8l^4 e^4 m M_4^2,
\end{aligned}
$$

where the last inequality assumes $\delta \leq \left(\frac{M_4^2}{96m(m+K)^{2l-2}d^{2l-4}}\right)^{1/4}$.

Denote $L_2 := \sqrt{20me^2\alpha^2 l^4 d^{l-1}(m+K)^{l-2} + 8l^4 e^4 m M_4^2}$. Then, we have

$$
f(U',\bar{C}') - f(U',\bar{C}) \leq \left\langle \nabla_{\hat{U}} f(U',\bar{C}), -\eta\nabla_U f(U,\bar{C})\right\rangle + \frac{\eta^2 L_2}{2}\left\|\nabla_U f(U,\bar{C})\right\|_F^2.
$$

From equation 3 and 5 we know that $\forall i\in[m]$, $\nabla_{\hat{u}_i} f(U,\bar{C}) = -\frac{l-2}{l}\cdot\frac{u_i u_i^\top(\nabla_{u_i}f(U,\bar{C}))}{\|u_i\|^2}$. Thus,

$$
\left\|\nabla_{\hat{U}} f(U,\bar{C})\right\|_F \leq \frac{l-2}{l}\left\|\nabla_U f(U,\bar{C})\right\|_F.
$$

In order to bound $\left\|\nabla_{\hat{U}} f(U',\bar{C})\right\|_F$, we still need to show that $\nabla_{\hat{U}} f(U',\bar{C})$ is close to $\nabla_{\hat{U}} f(U,\bar{C})$.

**Bounding** $\left\|\nabla_{\hat{U}} f(U',\bar{C}) - \nabla_{\hat{U}} f(U,\bar{C})\right\|_F$: Define $U''$ as $(1-\theta)U + \theta U'$ for all $0\leq\theta\leq 1$. We will show that the derivative of $\nabla_{\hat{U}} f$ in $U$ evaluated $(U'',\bar{C})$ is bounded. We denote this derivative as $\tilde{\mathcal{H}}''$ that is a $dm\times dm$ matrix. Matrix $\tilde{\mathcal{H}}''$ contains $m\times m$ blocks each of which has dimension $d\times d$ and corresponds to $\frac{\partial^2}{\partial\hat{u}_i\partial u_j}f(U'',\bar{C})$. Denote $T''$ as the tensor parameterized by $(U'',\bar{C})$. Recall that,

$$
\begin{aligned}
&\frac{\partial}{\partial\hat{u}_i} f(U'',\bar{C}) \\
&= -(l-2)a_i(T''-T^*)((u_i'')^{\otimes l})\frac{u_i}{\|u_i\|^l}\left(\left(\sqrt{d(m+K)}\right)^{l-2}\mathbb{1}_{c_i=\sqrt{d(m+K)}/\|u_i\|} + \mathbb{1}_{c_i=1/\|u_i\|}\right) \\
&\quad - \lambda(l-2)\frac{u_i}{\|u_i\|^l}\|u_i''\|^l.
\end{aligned}
$$

For any $i\in[m]$, we have

$$
\begin{aligned}
&\frac{\partial^2}{\partial u_i\partial\hat{u}_i} f(U'',\bar{C}) \\
&= -l(l-2)a_i(T''-T^*)((u_i'')^{\otimes l-1},I)\frac{u_i^\top}{\|u_i\|^l}\left(\left(\sqrt{d(m+K)}\right)^{l-2}\mathbb{1}_{c_i=\sqrt{d(m+K)}/\|u_i\|} + \mathbb{1}_{c_i=1/\|u_i\|}\right) \\
&\quad - l(l-2)c_i^{l-2}\|u_i''\|^{2l-2}\frac{u_i(u_i'')^\top}{\|u_i\|^l}\left(\left(\sqrt{d(m+K)}\right)^{l-2}\mathbb{1}_{c_i=\sqrt{d(m+K)}/\|u_i\|} + \mathbb{1}_{c_i=1/\|u_i\|}\right) \\
&\quad - \lambda l(l-2)\frac{u_i(u_i'')^\top}{\|u_i\|^l}\|u_i''\|^{l-2}.
\end{aligned}
$$

The Frobenius norm of $\frac{\partial^2}{\partial u_i \partial \hat{u}_i} f(U'', \bar{C})$ can be bounded as follows,

$$
\begin{aligned}
\left\| \frac{\partial^2}{\partial u_i \partial \hat{u}_i} f(U'', \bar{C}) \right\|_F &\leq \sqrt{e} l^2 \beta (\sqrt{d(m+K)})^{l-2} + \sqrt{e} l^2 \max\left( \|u_i\|^2, 4d^{l-2}(m+K)^{l-1}\delta^2 \right) + \sqrt{e}\lambda l^2 \\
&\leq 2\sqrt{e} l^2 \beta (\sqrt{d(m+K)})^{l-2} + \sqrt{e} l^2 \max\left( \|u_i\|^2, 4d^{l-2}(m+K)^{l-1}\delta^2 \right),
\end{aligned}
$$

where the last inequality assumes $\lambda \leq 1$. Therefore,

$$
\left\| \frac{\partial^2}{\partial u_i \partial \hat{u}_i} f(U'', \bar{C}) \right\|_F^2 \leq 8el^4 \beta^2 d^{l-2}(m+K)^{l-2} + 2el^4 \max\left( \|u_i\|^4, 16d^{2l-4}(m+K)^{2l-2}\delta^4 \right)
$$

For $i \neq j$, we have

$$
\begin{aligned}
&\frac{\partial^2}{\partial u_j \partial \hat{u}_i} f(U'', \bar{C}) \\
&= -l(l-2)a_i a_j c_j^{l-2} \langle u_i'', u_j'' \rangle^{l-1} \frac{u_i(u_i'')^\top}{\|u_i\|^l} \left( (\sqrt{d(m+K)})^{l-2} \mathbb{1}_{c_i = \sqrt{d(m+K)}/\|u_i\|} + \mathbb{1}_{c_i = 1/\|u_i\|} \right)
\end{aligned}
$$

The Frobenius norm square can be bounded as

$$
\begin{aligned}
\left\| \frac{\partial^2}{\partial \hat{u}_i \partial u_j} f(U'', C) \right\|_F^2 &\leq el^4 \max\{4d^{l-2}(m+K)^{l-1}\delta^2, \|u_i\|^2\} \max\{4d^{l-2}(m+K)^{l-1}\delta^2, \|u_j\|^2\} \\
&\leq el^4 \left( \|u_i\|^4 + \|u_j\|^4 + 16(m+K)^{2l-2}\delta^4 d^{2l-4} \right).
\end{aligned}
$$

Summing over the bounds on blocks, we can bound the Frobeneius norm of $\tilde{\mathcal{H}}''$,

$$
\begin{aligned}
&\left\| \tilde{\mathcal{H}}'' \right\|_F^2 \\
&= \sum_{i,j} \left\| \frac{\partial^2}{\partial u_j \partial \hat{u}_i} f(U'', \bar{C}) \right\|_F^2 \\
&\leq 8mel^4 \beta^2 d^{l-2}(m+K)^{l-2} + 2el^4 \sum_{i=1}^m \|u_i\|^4 + 32mel^4 d^{2l-4}(m+K)^{2l-2}\delta^4 \\
&\quad + 2mel^4 \sum_{i=1}^m \|u_i\|^4 + 16m^2 el^4 (m+K)^{2l-2}\delta^4 d^{2l-4} \\
&\leq 8mel^4 \beta^2 d^{l-2}(m+K)^{l-2} + 48m^2 el^4 d^{2l-4}(m+K)^{2l-2}\delta^4 + 3el^4 m M_4^2 \\
&\leq 8mel^4 \beta^2 d^{l-2}(m+K)^{l-2} + 4el^4 m M_4^2,
\end{aligned}
$$

where the last inequality assumes $\delta \leq \left( \frac{M_4^2}{48md^{2l-4}(m+K)^{2l-2}} \right)^{1/4}$.

Denoting $L_3 := \sqrt{8mel^4 \beta^2 d^{l-2}(m+K)^{l-2} + 4el^4 m M_4^2}$, we have

$$
\left\| \nabla_{\hat{U}} f(U', \bar{C}) - \nabla_{\hat{U}} f(U, \bar{C}) \right\|_F \leq L_3 \left\| \eta \nabla_U f(U, \bar{C}) \right\|_F \leq \frac{1}{3l} \left\| \nabla_U f(U, \bar{C}) \right\|_F,
$$

where the second inequality assumes $\eta \leq \frac{1}{3L_3 l}$. Therefore, we have

$$
\begin{aligned}
\left\| \nabla_{\hat{U}} f(U', \bar{C}) \right\|_F \leq \left\| \nabla_{\hat{U}} f(U, \bar{C}) \right\|_F + \frac{1}{3l} \left\| \nabla_U f(U, \bar{C}) \right\|_F &\leq \left( \frac{l-2}{l} + \frac{1}{3l} \right) \left\| \nabla_U f(U, \bar{C}) \right\|_F \\
&\leq \left( 1 - \frac{5}{3l} \right) \left\| \nabla_U f(U, \bar{C}) \right\|_F.
\end{aligned}
$$

Overall, we have proved that as long as $\eta$ is small enough,

$$
\begin{aligned}
f(U', \bar{C}') - f(U, \bar{C}) =& f(U', \bar{C}) - f(U, \bar{C}) + f(U', \bar{C}') - f(U', \bar{C}) \\
\leq& -\eta \left\| \nabla_U f(U, \bar{C}) \right\|_F^2 + \frac{\eta^2 L_1}{2} \left\| \nabla_U f(U, \bar{C}) \right\|_F^2 \\
&+ \eta \left\| \nabla_{\hat{U}} f(U', \bar{C}) \right\|_F^2 + \frac{\eta^2 L_2}{2} \left\| \nabla_{\hat{U}} f(U', \bar{C}) \right\|_F^2 \\
\leq& -\eta \left\| \nabla_U f(U, \bar{C}) \right\|_F^2 + \frac{\eta^2 L_1}{2} \left\| \nabla_U f(U, \bar{C}) \right\|_F^2 \\
&+ \eta \left( 1 - \frac{5}{3l} \right) \left\| \nabla_U f(U, \bar{C}) \right\|_F^2 + \frac{\eta^2 L_2}{2} \left\| \nabla_U f(U, \bar{C}) \right\|_F^2 \\
\leq& -\frac{\eta}{l} \left\| \nabla_U f(U, \bar{C}) \right\|_F^2,
\end{aligned}
$$

where the last inequality assumes $\eta \leq \frac{4}{3l(L_1+L_2)}$. Combining all the bounds on $\delta, \eta$, we know there exists constant $\mu_1, \mu_2$ such that as long as $\delta \leq \frac{\mu_1}{m^{\frac{1}{4}} \sqrt{\lambda} d^{\frac{l-2}{2}} (m+K)^{\frac{l-1}{2}}}, \eta \leq \frac{\mu_2 \lambda}{m^{\frac{1}{2}} d^{\frac{l-1}{2}} (m+K)^{\frac{l-2}{2}}}$, we have

$$
f(U', \bar{C}') - f(U, \bar{C}) \leq -\frac{\eta}{l} \left\| \nabla_U f(U, \bar{C}) \right\|_F^2.
$$

$\square$

Then, we know in an epoch, the function value can only increase because of the initialization and the scalar mode switches. In Lemma 15, we show these operations cannot increase the function by too much. Note that Lemma 15 is the formal version of Lemma 4 together with the bound for scalar mode switches in the main text (these two arguments in the main text correspond to the two claims in Lemma 15).

**Lemma 15.** *Assume $f(U_0', \bar{C}_0') \leq \tilde{\Gamma} \leq 10$ at the beginning of an epoch, $\delta \leq \frac{\mu_1 \sqrt{\tilde{\Gamma}}}{m^{\frac{3}{4}} d^{\frac{l-2}{2}} (m+K)^{\frac{l-1}{2}} \lambda^{\frac{1}{2}}}$, and $\eta \leq \frac{\mu_2 \lambda}{m^{\frac{1}{2}} d^{\frac{l-1}{2}} (m+K)^{\frac{l-2}{2}}}$ for some constants $\mu_1, \mu_2$. Also assume that $\lambda m \geq 10$. Denote the parameters at the end of this epoch as $(U_H, \bar{C}_H)$, then*

$$
f(U_H, \bar{C}_H) \leq \exp \left( \frac{14}{\lambda m} \right) \tilde{\Gamma}.
$$

**Proof of Lemma 15.** By Lemma 14, we know the function value does not increase in any iteration (before potential scalar mode switch) as long as the initial function value is at most 10 and $\delta \leq \frac{\mu_1}{m^{\frac{1}{4}} \sqrt{\lambda} d^{\frac{l-2}{2}} (m+K)^{\frac{l-1}{2}}}, \eta \leq \frac{\mu_2 \lambda}{m^{\frac{1}{2}} d^{\frac{l-1}{2}} (m+K)^{\frac{l-2}{2}}}$ for some constants $\mu_1, \mu_2$. Thus, the function value can only increase when we reinitialize a component or when we switch the scaling from $\sqrt{d(m+K)}/ \|u_i\|$ to $1/ \|u_i\|$. In the following, we first show that reinitializing a component can only increase the function value by a small factor.

**Claim 1.** *Suppose $f(U, \bar{C}) \leq \hat{\Gamma} \leq 10$. Reinitialize any vector with the smallest $\ell_2$ norm among all columns of $U$, and let the updated parameters be $(U', \bar{C}')$, then*

$$
f(U', \bar{C}') \leq \left( 1 + \frac{13}{\lambda m} \right) \hat{\Gamma}.
$$

According to the definition of the function value, we know $\|T - T^*\|_F \leq \sqrt{2\hat{\Gamma}} \leq \sqrt{20}$, $\sum_{j=1}^m \|u_j\|^2 \leq \frac{\hat{\Gamma}}{\lambda}$, and $\sum_{j=1}^m \|u_j\|^4 \leq \left( \sum_{j=1}^m \|u_j\|^2 \right)^2 \leq \frac{\hat{\Gamma}^2}{\lambda^2}$. Suppose $u_i$ is one of the vectors in $U$ with the smallest $\ell_2$ norm, then $\|u_i\|^2 \leq \frac{\hat{\Gamma}}{\lambda m}$. Suppose $u_i', c_i', \hat{c}_i', a_i'$ are the corresponding

reinitialized vector and coefficients, and we have

$$\left\| a_i c_i^{l-2} u_i^{\otimes l} - a_i'(c_i')^{l-2}(u_i')^{\otimes l} \right\|_F$$

$$\leq \left\| a_i c_i^{l-2} u_i^{\otimes l} \right\|_F + \left\| a_i'(c_i')^{l-2}(u_i')^{\otimes l} \right\|_F$$

$$\leq \max\left( \|u_i\|^2, (\sqrt{d(m+K)})^{l-2}4(m+K)\delta^2 \right) + (\sqrt{d(m+K)})^{l-2}\delta^2$$

$$\leq \frac{\hat{\Gamma}}{\lambda m} + (\sqrt{d(m+K)})^{l-2}4(m+K)\delta^2 + (\sqrt{d(m+K)})^{l-2}\delta^2 \leq \frac{2\hat{\Gamma}}{\lambda m},$$

where the last inequality assumes $\delta^2 \leq \frac{\hat{\Gamma}}{5\lambda m(m+K)^{\frac{l}{2}}d^{\frac{l-2}{2}}}$. Therefore, we can bound $f(U', \bar{C}')$ as

$$f(U', \bar{C}')$$

$$= \frac{1}{2}\left\| T - T^* - a_i c_i^{l-2} u_i^{\otimes l} + a_i'(c_i')^{l-2}(u_i')^{\otimes l} \right\|_F^2 + \lambda\sum_{j=1}^m \hat{c}_j^{l-2}\|u_j\|^l - \lambda\hat{c}_i^{l-2}\|u_i\|^{2l} + \lambda(\hat{c}_i')^{l-2}\|u_i'\|^l$$

$$\leq f(U,C) + \|T - T^*\|_F \left\| a_i c_i^{l-2}u_i^{\otimes l} - a_i'(c_i')^{l-2}(u_i')^{\otimes l} \right\|_F + \frac{1}{2}\left\| a_i c_i^{l-2}u_i^l - a_i'(c_i')^{l-2}(u_i')^l \right\|_F^2 + \lambda(\hat{c}_i')^{l-2}\|u_i'\|^l$$

$$\leq f(U,C) + \sqrt{20}\cdot\frac{2\hat{\Gamma}}{\lambda m} + \frac{1}{2}\left(2\frac{\hat{\Gamma}}{\lambda m}\right)^2 + \lambda\delta^2$$

$$\leq \left(1 + \frac{12}{\lambda m}\right)\hat{\Gamma} + \lambda\delta^2$$

$$\leq \left(1 + \frac{13}{\lambda m}\right)\hat{\Gamma},$$

where the second last inequality assumes $\lambda m \geq \hat{\Gamma}$ and the last inequality assumes $\delta^2 \leq \frac{\hat{\Gamma}}{\lambda^2 m}$.

Switching the scaling from $\sqrt{d(m+K)}/\|u_i\|$ to $1/\|u_i\|$ can also potentially increase the function value. In the following, we show that the function value increase is small because we only switch the scaling mode when $\|u_i\| \leq 2\sqrt{m+K}\delta$.

**Claim 2.** *Assume $f(U', \bar{C}') \leq \bar{\Gamma}$. Suppose at this iteration we switch the scaling of $c_i'$, i.e., we set $c_i'$ as $c_i'/\sqrt{d(m+K)}$. Let the updated parameters be $(U', C'', \hat{C}', A')$, we have*

$$f(U', C'', \hat{C}', A') \leq \left(1 + \frac{1}{\lambda m^2}\right)\bar{\Gamma}.$$

Suppose $u_i$ is the parameter which is one step of gradient descent before $u_i'$. According to the algorithm, we know $\|u_i\| \leq 2\sqrt{m+K}\delta$. According to the proof in Lemma 14 where we bound the derivative of $f$ with respect to $u_i$, we know that as long as $\eta \leq \frac{1}{l(\sqrt{2\Gamma}(\sqrt{d(m+K)})^{l-2}+\lambda)}$, we have $\|u_i'\| \leq 2\|u_i\| \leq 4\sqrt{m+K}\delta$. Therefore,

$$\left\| (c_i')^{l-2}(u_i')^{\otimes l} - (c_i'')^{l-2}(u_i')^{\otimes l} \right\|_F = \left\| (c_i')^{l-2}(u_i')^{\otimes l} - \frac{1}{(\sqrt{d(m+K)})^{l-2}}(c_i')^{l-2}(u_i')^{\otimes l} \right\|_F$$

$$\leq \left\| (c_i')^{l-2}(u_i')^{\otimes l} \right\|_F \leq 16(m+K)\delta^2(\sqrt{d(m+K)})^{l-2}.$$

Suppose the tensor at $(U', \bar{C}')$ is $T'$, then $\|T' - T^*\|_F \leq \sqrt{2\bar{\Gamma}}$.

Thus, we can bound $f(U', C'', \hat{C}', A')$ as follows:

$$f(U', C'', \hat{C}', A')$$

$$\leq f(U', \bar{C}'') + \|T' - T^*\|_F \left\| (c_i')^{l-2}(u_i')^{\otimes l} - (c_i'')^{l-2}(u_i')^{\otimes l} \right\|_F$$

$$\quad + \frac{1}{2}\left\| (c_i')^{l-2}(u_i')^{\otimes l} - (c_i'')^{l-2}(u_i')^{\otimes l} \right\|_F^2$$

$$\leq \bar{\Gamma} + \sqrt{2\bar{\Gamma}}\left(16(m+K)\delta^2(\sqrt{d(m+K)})^{l-2}\right) + \frac{1}{2}\left(16(m+K)\delta^2(\sqrt{d(m+K)})^{l-2}\right)^2$$

$$\leq \left(1 + \frac{1}{\lambda m^2}\right)\bar{\Gamma},$$

where the last inequality assumes $\delta^2 \leq \frac{\sqrt{\Gamma}}{32\sqrt{2}\lambda m^2(m+K)^{\frac{1}{2}}d^{\frac{l-2}{2}}}$ and $\lambda m^2 \geq 1$.

We are now ready to bound the increase of the function value during this epoch. According to the algorithm, each epoch contains at most $m$ scaling mode switches. Therefore, following Claim 2, all the scaling switches in one epoch can increase the upper bound of the function value by at most a factor of $\left(1 + \frac{1}{\lambda m^2}\right)^m \leq \exp\left(\frac{1}{\lambda m}\right)$. Combining with Claim 1 which considers the re-initialization, we know that in each epoch, the upper bound of the function value increase by at most a factor of $\exp\left(\frac{1}{\lambda m}\right)\left(1 + \frac{13}{\lambda m}\right) \leq \exp\left(\frac{14}{\lambda m}\right)$. $\qquad\square$

Following Lemma 14 and Lemma 15, we are ready to show that the function value is upper bounded by a constant by an induction proof. At the beginning of our algorithm, the function value is bounded by a constant as long as the initialization radius $\delta$ is small enough. According to Lemma 15, the increase in each epoch is bounded by a factor of $O(\frac{1}{\lambda m})$. Therefore, as long as the total number of epochs do not exceed $O(\lambda m)$, $f$ will always be bounded by a constant. As a consequence, the Frobenius norm square of $U$ must be bounded by $O(\frac{1}{\lambda})$ due to the design of the regularizer. These results are summarized into Lemma 16.

**Lemma 16.** *Assume* $\delta \leq \frac{\mu_1}{m^{\frac{3}{4}}d^{\frac{l-2}{2}}(m+K)^{\frac{l-1}{2}}\lambda^{\frac{1}{2}}}$, *and* $\eta \leq \frac{\mu_2\lambda}{m^{\frac{1}{2}}l^4 d^{\frac{l-1}{2}}(m+K)^{\frac{l-2}{2}}}$ *for some constants* $\mu_1, \mu_2$. *Also assume* $K \leq \frac{\lambda m}{14}$ *and* $\frac{10}{m} \leq \lambda \leq 1$. *We know throughout the algorithm*

$$f(U, \bar{C}) \leq 10 \text{ and } \sum_{i=1}^{m} \|u_i\|^2 \leq \frac{10}{\lambda}.$$

**Proof of Lemma 16.** Let's first show that the function value is bounded at the initialization if we choose $\delta$ to be small enough. At initialization, we have

$$
\begin{aligned}
f(U, \bar{C}) =& \frac{1}{2}\|T - T^*\|_F^2 + \lambda \sum_{i=1}^{m} \hat{c}_i^{l-2}\|u_i\|^l \\
\leq& \frac{1}{2}\left(\sum_{i=1}^{m}\|c_i^{l-2}u_i^{\otimes l}\|_F + \|T^*\|_F\right)^2 + \lambda \sum_{i=1}^{m}\|u_i\|^2 \\
\leq& \frac{1}{2}\left(m(\sqrt{d(m+K)})^{l-2}\delta^2 + 1\right)^2 + \lambda m \delta^2 \\
\leq& m^2 d^{l-2}(m+K)^{l-2}\delta^4 + 1 + \lambda m \delta^2 \leq 3,
\end{aligned}
$$

where the last inequality assumes $\delta^4 \leq \frac{1}{m^2 d^{l-2}(m+K)^{l-2}}$ and $\lambda \leq 1$.

We use an inductive proof to prove that the function value at the end of the $k$-th ($k \leq K$) iteration is at most $3\exp(\frac{14k}{\lambda m})$: At the initialization, the function value is at most 3. For every epoch, assume that our induction hypothesis is true, then at each step (a step can be a re-initialization, a gradient descent update, or a scalar mode switch), from Lemma 15 we know that the function value is upper bounded by $3\exp(\frac{14k}{\lambda m}) \leq 10$, so at this step Lemma 14 is correct, meaning that Lemma 15 is still correct at the next step.

Therefore, throughout the algorithm, we have

$$f(U, C) \leq 3\exp\left(\frac{14K}{\lambda m}\right) \leq 10,$$

where we assume $K \leq \frac{\lambda m}{14}$. This immediately implies that $\sum_i \|u_i\|^2 \leq \frac{10}{\lambda}$ by the design of our regularizer. $\qquad\square$

Now we are ready to prove Lemma 13.

**Proof of Lemma 13.** From Lemma 16, we know that the function value is upper bounded by 10 throughout our algorithm. Besides, from Lemma 15 we know that the function value increase is at most $(\exp(\frac{14}{\lambda m})-1)$ times the function value at the beginning of this epoch. Choosing $\tilde{\Gamma} = f(U_0', \bar{C}_0')$, we know the function value increase at each epoch cannot exceed $10(\exp(\frac{14}{\lambda m}) - 1) = O(\frac{1}{\lambda m})$, which finishes the proof of Lemma 13. $\qquad\square$

## C.2 Escaping local minima

In this section, we will give a formal proof of Lemma 12. We again follow the proof ideas outlined in Section 5.2. Recall that the proof goes in the following steps:

1. We first show that the projection of $U$ in the $B$ subspace must be very small, therefore the influence from incorrect subspace $B$ is small (Lemma 17).

2. We then focus on the correlation in the correct subspace $S$. First we show that the correlation can be significantly negative at re-initialization with constant probability (Lemma 18).

3. If the correlation is always significantly negative, then the re-initialized component will grow exponentially and eventually decrease the function value (Lemma 19).

4. If the correlation changes significantly, the function value must also decrease (Lemma 20 and Lemma 21).

First of all, we need to show that the influence coming from $B$ is small enough so that it can be ignored. The following lemma is the formal version of Lemma 5. Note that the assumption $\|U\|_F \leq \sqrt{\frac{10}{\lambda}}$ has been verified in Lemma 16 so Lemma 17 holds for the entire algorithm.

**Lemma 17.** *Assume $\|U\|_F \leq \sqrt{\frac{10}{\lambda}}$ throughout the algorithm. Assume $\lambda \leq \sqrt{10}, \delta \leq \frac{\sqrt{10}}{2\sqrt{\lambda m d^{l-2}(m+K)^{l-1}}}$ and $\eta \leq \frac{\lambda}{20l}$. Then, we know $\|P_B U\|_F^2 \leq (m+K)\delta^2$ throughout the algorithm.*

**Proof of Lemma 17.** At the initialization,

$$\|P_B U\|_F^2 \leq \|U\|_F^2 = m\delta^2.$$

At the beginning of each epoch, we re-initialize one column of $U$, which at most increases $\|P_B U\|_F^2$ by $\delta^2$. Thus, the total increase due to the re-initialization process is at most $K\delta^2$.

Then, we only need to show that running gradient descent does not increase the norm of $P_B U$. Suppose at the beginning of one iteration, the tensor $T$ is parameterized by $(U, \bar{C})$. Let $U'$ be the updated parameter, which means $U' = U - \eta \nabla_U f(U, \bar{C})$. We have,

$$
\begin{aligned}
&\|P_B U'\|_F^2 - \|P_B U\|_F^2 \\
&= \left\|P_B(U - \eta \nabla_U f(U, \bar{C}))\right\|_F^2 - \|P_B U\|_F^2 \\
&= -2\eta \left\langle P_B U, P_B \nabla_U f(U, \bar{C}) \right\rangle + \eta^2 \left\|P_B \nabla_U f(U, \bar{C})\right\|_F^2.
\end{aligned}
\tag{6}
$$

We will show that the first term is negative and dominates the second term when $\eta$ is small enough, which implies that gradient descent never increases the norm of $P_B U$. We first compute the gradient as follows,

$$\nabla_U f(U, \bar{C}) = l\,\mathrm{mat}(T - T^*)U^{\odot l-1}C^{l-2}A + l\lambda U.$$

where $\mathrm{mat}(T - T^*) = UC^{l-2}A(U^{\odot l-1})^\top - U^* C^*[(U^*)^{\odot l-1}]^\top$ is a $d \times d^{l-1}$ matrix. Therefore, the projection of the gradient on $B$ subspace is

$$P_B \nabla_U f(U, \bar{C}) = lP_B\mathrm{mat}(T)U^{\odot l-1}C^{l-2}A + l\lambda P_B U.$$

Now, we show that the first term in (6) is negative.

$$
\begin{aligned}
-2\eta \left\langle P_B U, P_B \nabla_U f(U, \bar{C}) \right\rangle =& -2l\eta \left\langle P_B U, P_B\mathrm{mat}(T)U^{\odot l-1}C^{l-2}A + \lambda P_B U \right\rangle \\
=& -2l\eta \left\langle P_B U C^{l-2}A[U^{\odot l-1}]^\top, P_B\mathrm{mat}(T) \right\rangle - 2l\eta \left\langle P_B U, \lambda P_B U \right\rangle \\
=& -2l\eta \left\|P_B\mathrm{mat}(T)\right\|_F^2 - 2l\lambda\eta \left\|P_B U\right\|_F^2.
\end{aligned}
$$

Next, we show the second term in (6) is bounded. We have,

$$
\begin{aligned}
\eta^2 \left\|P_B \nabla_U f(U, \bar{C})\right\|_F^2 =& \eta^2 \left\|lP_B\mathrm{mat}(T)U^{\odot l-1}C^{l-2}A + l\lambda P_B U\right\|_F^2 \\
\leq& 2\eta^2 l^2 \left( \left\|P_B\mathrm{mat}(T)U^{\odot l-1}C^{l-2}A\right\|_F^2 + \lambda^2 \left\|P_B U\right\|_F^2 \right)
\end{aligned}
$$

Recall that $M_2 = \sqrt{\frac{10}{\lambda}}$. Note that

$$\left\| P_B \text{mat}(T) U^{\odot l-1} C^{l-2} A \right\|_F^2 \le \left\| P_B \text{mat}(T) \right\|_F^2 \left\| U^{\odot l-1} C^{l-2} \right\|_F^2$$

$$= \left\| P_B \text{mat}(T) \right\|_F^2 \sum_{i=1}^m c_i^{2l-4} \| u_i \|^{2l-2}$$

$$\le \left\| P_B \text{mat}(T) \right\|_F^2 \sum_{i=1}^m \max\{ 4(d(m+K))^{l-2}(m+K)\delta^2, \| u_i \|^2 \}$$

$$\le M_2^2 \left\| P_B \text{mat}(T) \right\|_F^2$$

where the second inequality holds since $c_i = \frac{\sqrt{d(m+K)}}{\| u_i \|}$ only when $\| u_i \| \le 2\sqrt{m+K}\delta$, and otherwise $c_i = \frac{1}{\| u_i \|}$. The last inequality assumes $\delta \le \frac{M_2}{2\sqrt{m d^{l-2}(m+K)^{l-1}}}$.

Overall, we have

$$\| P_B U' \|_F^2 - \| P_B U \|_F^2$$

$$= -2\eta \langle P_B U, P_B \nabla_U f(U, \bar{C}) \rangle + \eta^2 \left\| P_B \nabla_U f(U, \bar{C}) \right\|_F^2$$

$$\le -2l\eta \left( \left\| P_B \text{mat}(T) \right\|_F^2 + \lambda \| P_B U \|_F^2 \right) + 2\eta^2 l^2 \left( M_2^2 \left\| P_B \text{mat}(T) \right\|_F^2 + \lambda^2 \| P_B U \|_F^2 \right)$$

$$\le -l\eta \left( \left\| P_B \text{mat}(T) \right\|_F^2 + \lambda \| P_B U \|_F^2 \right),$$

where the last inequality assumes $\lambda \le M_2^2$ and $\eta \le \frac{1}{2l M_2^2}$. $\qquad \square$

Lemma 17 shows that the norm of $P_B U$ only increases at the (re-)initializations, so it will stay small throughout this algorithm. This allows us to bound the influence to our algorithm from the orthogonal subspace $B$ and only focus on subspace $S$. We denote the re-initialized vector at $t$-th step as $u_t$, and its sign as $a \in \{\pm 1\}$. Our analysis focuses on the correlation between $P_S u_t$ and the residual tensor

$$\langle P_{S^{\otimes l}} T_t - T^*, a \overline{P_S u_t}^{\otimes l} \rangle.$$

Here $\overline{P_S u_t}$ is the normalized version $P_S u_t$. We will show that if this correlation is significantly negative at every iteration the norm of $u_t$ will blow up exponentially.

Towards this goal, first we will show that the initial point $P_S u_0$ has a large negative correlation with the residual. We will lower bound this correlation by anti-concentration of Gaussian polynomials, and the following lemma is the formal version of Lemma 6. Note in our notation, we have $\langle P_{S^{\otimes l}} T_t - T^*, a \overline{P_S u_t}^{\otimes l} \rangle = a \left( P_{S^{\otimes l}} T_t - T^* \right) \left( \overline{P_S u_t}^{\otimes l} \right)$.

**Lemma 18.** *Suppose the residual at the beginning of one epoch is $T_0' - T^*$. Suppose $a c_0^{l-2} u_0^{\otimes l}$ is the reinitialized component. There exists absolute constant $\mu$ such that with probability at least $1/5$,*

$$\left\langle P_{S^{\otimes l}} T_0' - T^*, a \overline{P_S u_0}^{\otimes l} \right\rangle \le -\frac{1}{(\mu r l)^{l/2}} \left\| P_{S^{\otimes l}} T_0' - P_{S^{\otimes l}} T^* \right\|_F,$$

*where $\overline{P_S u_0} = P_S u_0 / \| P_S u_0 \|$.*

**Proof of Lemma 18.** Let's restrict into the $r^l$-dimensional space $S^{\otimes l}$, and let $P_{S^{\otimes l}}$ be the projection operator that projects a $d^l$-dimensional tensor to the $r^l$-dimensional space $S^{\otimes l}$. Then, we can think of $(P_{S^{\otimes l}} T - P_{S^{\otimes l}} T^*)$ as an $r^l$ dimensional vector, and $\overline{P_S u}$ comes from uniform distribution on $\mathbb{S}^{r-1}$. Let $v$ be an $r$-dimensional standard normal vector, then $a(P_{S^{\otimes l}} T - P_{S^{\otimes l}} T^*)(\overline{P_S u}^{\otimes l})$ has the same distribution as $a(P_{S^{\otimes l}} T - P_{S^{\otimes l}} T^*)(v^l) \frac{1}{\| v \|^l}$.

Let's first show that the variance of $a(P_{S^{\otimes l}} T - P_{S^{\otimes l}} T^*)(v^l)$ is large:

$$\text{Var} \left[ a(P_{S^{\otimes l}} T - P_{S^{\otimes l}} T^*)(v^l) \right] = \mathbb{E} \left| a(P_{S^{\otimes l}} T - P_{S^{\otimes l}} T^*)(v^l) \right|^2$$

$$\ge l! \left\| P_{S^{\otimes l}} T - P_{S^{\otimes l}} T^* \right\|_F^2,$$

where the equality holds because $\mathbb{E}\left[a(P_{S^{\otimes l}}T - P_{S^{\otimes l}}T^*)(v^l)\right] = 0$ and the inequality follows from Lemma 11. It's not hard to see that $a(P_{S^{\otimes l}}T - P_{S^{\otimes l}}T^*)(v^l)$ is an $l$-th order polynomial of standard Gaussian vectors. By anti-concentration inequality of Gaussian polynomials (Lemma 25), we know there exists constant $\kappa$ such that

$$\Pr\left[\left|a(P_{S^{\otimes l}}T - P_{S^{\otimes l}}T^*)(v^l)\right| \leq \epsilon\sqrt{l!}\,\|P_{S^{\otimes l}}T - P_{S^{\otimes l}}T^*\|_F\right] \leq \kappa l \epsilon^{1/l}.$$

Choosing $\epsilon = \frac{1}{2^l \kappa^l l^l}$, we know with probability at least half,

$$\begin{aligned}
\left|a(P_{S^{\otimes l}}T - P_{S^{\otimes l}}T^*)(v^l)\right| &\geq \frac{1}{2^l \kappa^l l^l}\sqrt{l!}\,\|P_{S^{\otimes l}}T - P_{S^{\otimes l}}T^*\|_F \\
&\geq \frac{1}{2^l \kappa^l l^l}\left(\frac{l}{e}\right)^{l/2}\|P_{S^{\otimes l}}T - P_{S^{\otimes l}}T^*\|_F \\
&= \frac{1}{2^l e^{l/2}\kappa^l l^{l/2}}\|P_{S^{\otimes l}}T - P_{S^{\otimes l}}T^*\|_F\,.
\end{aligned}$$

Since the distribution of $a(P_{S^{\otimes l}}T - P_{S^{\otimes l}}T^*)(v^l)$ is symmetric, we know with probability at least $1/4$,

$$a(P_{S^{\otimes l}}T - P_{S^{\otimes l}}T^*)(v^l) \leq -\frac{1}{2^l e^{l/2}\kappa^l l^{l/2}}\|P_{S^{\otimes l}}T - P_{S^{\otimes l}}T^*\|_F\,.$$

According to Lemma 24, we know that with probability at least $19/20$,

$$\|v\| \leq \kappa'\sqrt{r},$$

where $\kappa'$ is some constant. This further implies that with probability at least $1/5$,

$$a(P_{S^{\otimes l}}T - P_{S^{\otimes l}}T^*)(v^l)\frac{1}{\|v\|^l} \leq -\frac{1}{2^l e^{l/2}(\kappa\kappa')^l(rl)^{l/2}}\|P_{S^{\otimes l}}T - P_{S^{\otimes l}}T^*\|_F\,.$$

Choosing $\mu = 4e\kappa^2(\kappa')^2$ finishes the proof. $\qquad\square$

Our next step argues that if this negative correlation is large in every step, then the norm of $u_t$ blows up exponentially. Intuitively, this is due to the fact that the correlation is basically the dominating term in the gradient, so when it is significantly negative the vector $u_t$ behaves similar to a vector doing matrix power method (here it is important that our model is 2-homogeneous so the behavior of power method is similar to the matrix setting). Below is the formal version of Lemma 7.

**Lemma 19.** *In the setting of Theorem 5, within one epoch, let $T_0$ be the tensor after the reinitilization and let $T_\tau$ be the tensor at the end of the $\tau$-th iteration. Assume $\|P_S u_0\| \geq \frac{\mu_2 \delta}{\sqrt{d}}$ for some constant $\mu_2 \in (0,1)$. For any $H \geq t \geq 1$, as long as $\left\langle P_{S^{\otimes l}}T_\tau - T^*, a\overline{P_S u_\tau}^{\otimes l}\right\rangle \leq \frac{-\epsilon}{5(\mu_1 rl)^{l/2}}$ for some constant $\mu_1$ for all $t-1 \geq \tau \geq 0$, we have*

$$\|P_S u_t\|^2 \geq \left(1 + \eta\left(\frac{\mu_2}{2}\right)^{l-2}\frac{\epsilon}{10(\mu_1 rl)^{\frac{l}{2}}}\right)^t \|P_S u_0\|^2\,.$$

**Proof of Lemma 19.** We will use inductive proof for this lemma. At the first step, we have

$$\begin{aligned}
\|P_S u_1\|^2 &= \left\|P_S u_0 - \eta P_S \nabla_u f(U_0, \bar{C}_0)\right\|^2 \\
&= \|P_S u_0\|^2 - \eta\left\langle P_S u_0, P_S \nabla_u f(U_0, \bar{C}_0)\right\rangle + \eta^2\left\|P_S \nabla_u f(U_0, \bar{C}_0)\right\|^2 \\
&\geq \|P_S u_0\|^2 - \eta\left\langle P_S u_0, P_S \nabla_u f(U_0, \bar{C}_0)\right\rangle\,.
\end{aligned}$$

We can write down the $P_S \nabla_u f(U_0, \bar{C}_0)$ as follows,

$$P_S \nabla_u f(U_0, \bar{C}_0) = al(T_0 - T^*)(u_0^{\otimes(l-1)}, P_S)c_0^{l-2} + \lambda l P_S u_0.$$

Let's first consider $al(T_0 - T^*)((u_0)^{\otimes(l-1)}, P_S)c_0^{l-2}$. We can decompose $u_0$ into $P_S u_0$ and $P_B u_0$, so we can divide $al(T_0 - T^*)(u_0^{\otimes(l-1)}, P_S)c_0^{l-2}$ into $2^{l-1}$ terms, each of which corresponds to

the projection of $u_0^{\otimes(l-1)}$ on a subspace in $\{S, B\}^{\otimes l-1}$. For subspace $S^{\otimes(l-1)}$, the projection is $al(P_{S^{\otimes l}}T_0 - P_{S^{\otimes l}}T^*)((P_S u_0)^{\otimes l-1}, P_S)c_0^{l-2}$. Its inner product with $-P_S u_0$ is

$$\left\langle -P_S u_0, al(P_{S^{\otimes l}}T_0 - P_{S^{\otimes l}}T^*)((P_S u_0)^{\otimes l-1}, P_S)c_0^{l-2} \right\rangle$$

$$= -al(P_{S^{\otimes l}}T_0 - P_{S^{\otimes l}}T^*)((P_S u_0)^{\otimes l})c_0^{l-2}$$

$$= -al(P_{S^{\otimes l}}T_0 - P_{S^{\otimes l}}T^*)((\overline{P_S u_0})^{\otimes l})\left(\|P_S u_0\|\, c_0\right)^{l-2}\|P_S u_0\|^2 .$$

Now, we only need to show that $\|P_S u_0\|\, c_0$ is lower bounded. We have

$$\|P_S u_0\|\, c_0 = \frac{\sqrt{d(m+K)}\,\|P_S u_0\|}{\|u_0\|} \geq \frac{\sqrt{d(m+K)}\mu_2\delta/\sqrt{d}}{\delta} \geq \frac{\mu_2}{2},$$

where the first inequality uses $\|P_S u_0\| \geq \mu_2\delta/\sqrt{d}$. Therefore,

$$\left\langle -P_S u_0, al(P_{S^{\otimes l}}T_0 - P_{S^{\otimes l}}T^*)((P_S u_0)^{\otimes l-1}, P_S)c_0^{l-2} \right\rangle \geq \left(\frac{\mu_2}{2}\right)^{l-2}\frac{\epsilon}{5(\mu_1 rl)^{l/2}}\|P_S u_0\|^2 .$$

We then bound the remaining terms in $al(T_0 - T^*)((u_0)^{\otimes l-1}, P_S)c_0^{l-2}$: For any $l-1 \geq k \geq 1$, we consider the subspace $B^{\otimes k} \otimes S^{\otimes l-1-k}$ and all of its permutations, we bound the norm of $al(T_0 - T^*)((P_B u_0)^{\otimes k}, (P_S u_0)^{\otimes l-1-k}, P_S)c_0^{l-2}$ as follows.

$$\left\|al(T_0 - T^*)\left((P_B u_0)^{\otimes k}, (P_S u_0)^{\otimes l-1-k}, P_S\right)c_0^{l-2}\right\|$$

$$= l\left\|T_0((P_B u_0)^{\otimes k}, (P_S u_0)^{\otimes l-1-k}, P_S)c_0^{l-2}\right\|$$

$$\leq l\sum_{i=1}^m c_{0,i}^{l-2}\|P_B u_{0,i}\|^k\|P_S u_{0,i}\|^{l-k}\|P_B u_0\|^k\|P_S u_0\|^{l-1-k}c_0^{l-2}$$

$$\leq l\sum_{i=1}^m c_{0,i}^{l-2}\|P_B u_{0,i}\|\,\|u_{0,i}\|^{l-1}\|P_B u_0\|\,\|u_0\|^{l-2}c_0^{l-2}$$

$$\leq l\sum_{i=1}^m d^{l-2}(m+K)^{l-1}\delta^2\|u_{0,i}\|$$

$$\leq l\sqrt{m}d^{l-2}(m+K)^{l-1}\delta^2 M_2,$$

where $M_2 = \sqrt{\frac{10}{\lambda}}$ is the upper bound of $\|U\|_F$. Denote $R_0$ as the summation of terms in all subspaces except for $S^{\otimes l-1}$. We have $\|R_0\| \leq (2^{l-1} - 1)l\sqrt{m}d^{l-2}(m+K)^{l-1}\delta^2 M_2$. Therefore, we have

$$|\langle P_S u_0, R_0\rangle| \leq \|P_S u_0\|\,\|R_0\| \leq 2^l l\sqrt{m}d^{l-2}(m+K)^{l-1}\delta^2 M_2\|P_S u_0\|$$

$$\leq \left(\frac{\mu_2}{2}\right)^{l-2}\frac{\epsilon}{20(\mu_1 rl)^{l/2}}\|P_S u_0\|^2$$

where the last inequality uses $\|P_S u_0\| \geq \frac{\mu_2\delta}{\sqrt{d}}$ and assumes $\delta \leq \frac{1}{2^l l\sqrt{m}d^{l-2}(m+K)^{l-1}M_2} \cdot \frac{\mu_2}{\sqrt{d}}\left(\frac{\mu_2}{2}\right)^{l-2}\frac{\epsilon}{20(\mu_1 rl)^{l/2}}$.

Next, let's analyze the regularizer $\lambda l P_S u_0$. Its norm can be bounded as follows,

$$\|\lambda l P_S u_0\| \leq \left(\frac{\mu_2}{2}\right)^{l-2}\frac{\epsilon}{20(\mu_1 rl)^{l/2}}\|P_S u_0\|,$$

where we assume $\lambda \leq \frac{1}{l} \cdot \left(\frac{\mu_2}{2}\right)^{l-2}\frac{\epsilon}{20(\mu_1 rl)^{l/2}}$.

Overall, we have

$$\|P_S u_1\|^2 \geq \|P_S u_0\|^2 - \eta\left\langle P_S u_0, \nabla_u f(U_0, \bar{C}_0)\right\rangle$$

$$\geq \|P_S u_0\|^2 + \eta\left(\frac{\mu_2}{2}\right)^{l-2}\left(\frac{\epsilon}{5(\mu_1 rl)^{l/2}} - \frac{\epsilon}{20(\mu_1 rl)^{l/2}} - \frac{\epsilon}{20(\mu_1 rl)^{l/2}}\right)\|P_S u_0\|^2$$

$$= \left(1 + \eta\left(\frac{\mu_2}{2}\right)^{l-2}\frac{\epsilon}{10(\mu_1 rl)^{l/2}}\right)\|P_S u_0\|^2 .$$

**Induction Step:** Suppose $\|P_S u_t\|^2 \geq \left(1 + \eta \left(\frac{\mu_2}{2}\right)^{l-2} \frac{\epsilon}{10(\mu_1 rl)^{l/2}}\right)^t \|P_S u_0\|^2$, we will prove that $\|P_S u_{t+1}\|^2 \geq \left(1 + \eta \left(\frac{\mu_2}{2}\right)^{l-2} \frac{\epsilon}{10(\mu_1 rl)^{l/2}}\right)^{t+1} \|P_S u_0\|^2$. Actually, we have assumed that $a(P_{S\otimes l}T_t - P_{S\otimes l}T^*)(\overline{P_S u_t}^{\otimes l}) \leq -\frac{\epsilon}{5(\mu_1 rl)^{l/2}}$, so we only need to show that $c_t \|P_S u_t\| \geq \frac{\mu_2}{2}$. Based on these two properties, the remaining proofs are exactly the same as that for $t = 0$.

The latter property is not hard to verify:

If $\|u_t\| > 2\sqrt{m+K}\delta$, we know $c_t = 1/\|u_t\|$. Then, we have $\|P_S u_t\| c_t = \frac{\|P_S u_t\|}{\|u_t\|} \geq \frac{\|u_t\| - \|P_B u_t\|}{\|u_t\|} \geq \frac{1}{2}$, where we use $\|P_B u_t\| \leq \sqrt{m+K}\delta$.

If $\|u_t\| \leq 2\sqrt{m+K}\delta$, we do not necessarily have $c_t = \frac{\sqrt{d(m+K)}}{\|u_t\|}$ because the norm of $\|u_t\|$ might first exceed the threshold and then drop below the threshold later. Note, by the induction proof, we only know the norm of $P_S u_t$ monotonically increase, which does not imply that $\|u_t\|$ monotonically increases. So, we have to consider both cases here. If $c_t = \frac{\sqrt{d(m+K)}}{\|u_t\|}$, we have $\|P_S u_t\| c_t = \frac{\sqrt{d(m+K)}\|P_S u_t\|}{\|u_t\|} \geq \frac{\mu_2}{2}$, which is because $\|P_S u_t\| \geq \|P_S u_0\| \geq \mu_2\delta/\sqrt{d}$. If $c_t = 1/\|u_t\|$, we know there exists $\tau \leq t$ such that $\|u_\tau\| > 2\sqrt{m+K}\delta$. Since $\|P_B u_\tau\| \leq \sqrt{m+K}\delta$, we know $\|P_S u_\tau\| \geq \sqrt{m+K}\delta$. By the induction proof, we know $\|P_S u_t\| \geq \|P_S u_\tau\| \geq \sqrt{m+K}\delta$. Then, we have $\|P_S u_t\| c_t = \frac{\|P_S u_t\|}{\|u_t\|} \geq \frac{1}{2}$.

This finishes the proof of Lemma 19. $\qquad\qquad\qquad\qquad\qquad\qquad\qquad\qquad\square$

Therefore the final step is to show that $a\overline{P_S u_t}^{\otimes l}$ always has a large negative correlation with $P_{S\otimes l}T_t - P_{S\otimes l}T^*$, unless the function value has already decreased. The difficulty here is that both the current reinitialized component $u_t$ and other components are moving, therefore $T_t$ is also changing.

We can bound the change of $T - T^*$ by separating it into two terms, which are the change of the re-initialized component and the change of the residual:

$$
\left| a(P_{S\otimes l}T_t - P_{S\otimes l}T^*)(\overline{P_S u_t}^{\otimes l}) - a(P_{S\otimes l}T_0 - P_{S\otimes l}T^*)(\overline{P_S u_0}^{\otimes l}) \right|
$$

$$
\leq \left| \sum_{\tau=1}^{t} \left( (P_{S\otimes l}T_{\tau-1} - P_{S\otimes l}T^*)(\overline{P_S u_\tau}^{\otimes l}) - (P_{S\otimes l}T_{\tau-1} - P_{S\otimes l}T^*)(\overline{P_S u_{\tau-1}}^{\otimes l}) \right) \right|
$$

$$
+ \sum_{\tau=1}^{t} \|T_\tau - T_{\tau-1}\|_F .
$$

The change of the re-initialized component has a small effect on the correlation because the change in $S$ subspace can only improve the correlation, and the influence of the $B$ subspace can be bounded. This is formally proved in the following lemma, which is the formal version of Lemma 8.

**Lemma 20.** *Assume* $\delta \leq \frac{\mu_1}{m^{\frac{3}{4}}\sqrt{\lambda} d^{\frac{l-2}{2}}(m+K)^{\frac{l-1}{2}}}, \eta \leq \frac{\mu_2}{\lambda^{\frac{3}{2}} m^{\frac{9}{4}} d^{\frac{3l-6}{2}}(m+K)^{\frac{3l-3}{2}}}$ *for some constants* $\mu_1, \mu_2$. *Assume* $K \leq \frac{\lambda m}{14}$ *and* $\frac{10}{m} \leq \lambda \leq 1$. *Suppose at the beginning of one iteration, the tensor $T$ is parameterized by $(U, \bar{C})$. Suppose $u$ is one column vector in $U$ with $\|P_S u\| \geq \frac{\mu_3 \delta}{\sqrt{d}}$ where $\mu_3$ is a constant. Suppose $u'$ is $u$ after one step of gradient descent: $u' = u - \eta \nabla_u f(U, \bar{C})$. We have*

$$
a(P_{S\otimes l}T - P_{S\otimes l}T^*)(\overline{P_S u'}^{\otimes l}) \leq a(P_{S\otimes l}T - P_{S\otimes l}T^*)(\overline{P_S u}^{\otimes l}) + \mu l^4 2^l d^{l-1.5} m^{1/2}(m+K)^{l-1}\eta\delta\lambda,
$$

*where $\mu$ is some constant.*

**Proof of Lemma 20.** Define $g(u) := a(P_{S\otimes l}T - P_{S\otimes l}T^*)(\overline{P_S u}^{\otimes l})$. Note that in function $g(u)$, we view $T$ as fixed. We will show that the change of $g$ is bounded when the input changes from $u$ to $u'$.

**Bounding first order change:**    Let's first compute the gradient of $g$ at $u$.

$$\nabla g(u) = al(P_{S\otimes l}T - P_{S\otimes l}T^*)((P_S u)^{\otimes l-1}, P_S)\frac{1}{\|P_S u\|^l} - al(P_{S\otimes l}T - P_{S\otimes l}T^*)((P_S u)^{\otimes l})\frac{P_S u}{\|P_S u\|^{l+2}}$$

$$= al(P_{S\otimes l}T - P_{S\otimes l}T^*)((P_S u)^{\otimes l-1}, P_S)\frac{1}{\|P_S u\|^l} - al(P_{S\otimes l}T - P_{S\otimes l}T^*)((P_S u)^{\otimes l-1}, \overline{P_S u})\frac{\overline{P_S u}}{\|P_S u\|^l}$$

$$= al(P_{S\otimes l}T - P_{S\otimes l}T^*)((P_S u)^{\otimes l-1}, P_S - \overline{P_S u}\cdot\overline{P_S u}^\top)\frac{1}{\|P_S u\|^l}$$

$$= al(P_{S\otimes l}T - P_{S\otimes l}T^*)((P_S u)^{\otimes l-1}, P_D)\frac{1}{\|P_S u\|^l},$$

where $P_D$ is the projection matrix on the span of $S \setminus \{u\}$. We can also compute the projection of $\nabla_u f(U, \bar{C})$ on $D$ as follows,

$$P_D \nabla_u f(U, \bar{C}) = al(T - T^*)(u^{\otimes l-1}, P_D)c^{l-2}.$$

We can divide $l(T - T^*)(u^{\otimes l-1}, P_D)c^{l-2}$ into $2^{l-1}$ terms, each of which corresponds to the projection of $u^{l-1}$ on a subspace. For subspace $S^{\otimes l-1}$, we have

$$al(T - T^*)((P_S u)^{\otimes l-1}, P_D)c^{l-2} = al(P_{S\otimes l}T - P_{S\otimes l}T^*)((P_S u)^{\otimes l-1}, P_D)c^{l-2},$$

which has non-negative inner product with $\nabla g(u)$. We can bound the norm of all the other terms. For any $l - 1 \geq k \geq 1$, consider subspace $B^{\otimes k} \otimes S^{\otimes(l-1-k)}$, we can bound the norm of $al(T - T^*)((P_B u)^{\otimes k}, (P_S u)^{\otimes(l-1-k)}, P_D)c^{l-2}$ as follows:

$$\left\| al(T - T^*)((P_B u)^{\otimes k}, (P_S u)^{\otimes l-1-k}, P_D)c^{l-2} \right\|$$

$$= l \left\| T((P_B u)^{\otimes k}, (P_S u)^{\otimes l-1-k}, P_D)c^{l-2} \right\|$$

$$\leq l \sum_{i=1}^m c_i^{l-2} \|P_B u_i\|^k \|P_S u_i\|^{l-k} \|P_B u\|^k \|P_S u\|^{l-1-k} c^{l-2}$$

$$\leq l \sum_{i=1}^m c_i^{l-2} \|P_B u_i\| \|u_i\|^{l-1} \|P_B u\| \|u\|^{l-2} c^{l-2}$$

$$\leq l \sum_{i=1}^m d^{l-2}(m + K)^{l-1}\delta^2 \|u_i\|$$

$$\leq l\sqrt{m}d^{l-2}(m + K)^{l-1}\delta^2 M_2,$$

where the second last inequality comes from Lemma 17.

Denote $R$ as the summation of terms in all subspaces except for $S^{\otimes l-1}$, then

$$\|R\| \leq (2^{l-1} - 1)l\sqrt{m}d^{l-2}(m + K)^{l-1}\delta^2 M_2.$$

Therefore, the first order change of $g$ can be bounded as follows,

$$\langle \nabla g(u), -\eta \nabla_u f(U, \bar{C}) \rangle$$

$$= \left\langle al(P_{S\otimes l}T - P_{S\otimes l}T^*)((P_S u)^{\otimes l-1}, P_D)\frac{1}{\|P_S u\|^l}, -\eta al(P_{S\otimes l}T - P_{S\otimes l}T^*)((P_S u)^{\otimes l-1}, P_D)c_u^{l-2} - \eta R \right\rangle$$

$$\leq \eta \left\| l(P_{S\otimes l}T - P_{S\otimes l}T^*)((P_S u)^{\otimes l-1}, P_D)\frac{1}{\|P_S u\|^l} \right\| \|R\|$$

$$\leq \eta l\sqrt{20}\frac{\sqrt{d}}{\mu_3\delta} \cdot (2^{l-1} - 1)l\sqrt{m}d^{l-2}(m + K)^{l-1}\delta^2 M_2$$

$$\leq \frac{10l^2 2^l}{\mu_3}\eta d^{l-1.5}\sqrt{m}(m + K)^{l-1}\delta M_2,$$

where the second last inequality assumes $\|P_S u\| \geq \frac{\mu_3\delta}{\sqrt{d}}$.

**Bounding higher order change:** For all $u'' = (1-\theta)u + \theta u'$ with $0 \le \theta \le 1$, we prove a uniform upper bound for $\left\|\nabla^2 g(u'')\right\|_F$. Recall the gradient of $g$ at $u''$,

$$\nabla g(u'') = al(P_{S^{\otimes l}}T - P_{S^{\otimes l}}T^*)((P_S u'')^{\otimes l-1}, P_D'')\frac{1}{\|P_S u''\|^l},$$

where $P_D''$ is the projection matrix to $S \setminus \{u''\}$. We compute $\left\|\nabla^2 g(u'')\right\|$ as follows,

$$\nabla^2 g(u'') = al(l-1)(P_{S^{\otimes l}}T - P_{S^{\otimes l}}T^*)((P_S u'')^{\otimes l-2}, P_S, P_D'')\frac{1}{\|P_S u''\|^l}$$

$$- al^2(P_{S^{\otimes l}}T - P_{S^{\otimes l}}T^*)((P_S u'')^{\otimes l-1}, P_D'') \otimes \frac{P_S u''}{\|P_S u''\|^{l+2}}.$$

Therefore,

$$\left\|\nabla^2 g(u'')\right\|_F \le 2l^2\sqrt{20}\frac{1}{\|P_S u''\|^2}.$$

Assume that $\eta \le \frac{\mu_3\delta\sqrt{\lambda}}{2\sqrt{10d}\left(\sqrt{20}l(\sqrt{d(m+K)})^{l-2}+\lambda l\right)}$ and from the proof of Lemma 14 where we bound the gradient, we know that

$$\left\|\eta\nabla_u f(U,\bar{C})\right\| \le \eta\left(l\sqrt{20}(\sqrt{d(m+K)})^{l-2} + \lambda l\right)\|u\| \le \frac{\sqrt{\frac{\lambda}{10}}\mu_3\delta}{2\sqrt{d}}\sqrt{\frac{10}{\lambda}} = \frac{\mu_3\delta}{2\sqrt{d}}.$$

Thus,

$$\|P_S u''\| \ge \|P_S u\| - \|P_S u'' - P_S u\|$$

$$\ge \|P_S u\| - \left\|\theta\eta\nabla_u f(U,\bar{C})\right\|$$

$$\ge \frac{\mu_3\delta}{\sqrt{d}} - \frac{\mu_3\delta}{2\sqrt{d}} = \frac{\mu_3\delta}{2\sqrt{d}}.$$

Therefore,

$$\left\|\nabla^2 g(u'')\right\|_F \le 2l^2\sqrt{20}\frac{4d}{\mu_3^2\delta^2}.$$

Overall, we have

$$g(u') - g(u) \le \left\langle\nabla g(u), -\eta\nabla_u f(U,\bar{C})\right\rangle + \frac{\eta^2}{2}2l^2\sqrt{20}\frac{4d}{\mu_3^2\delta^2}\left\|\nabla_u f(U,\bar{C})\right\|^2.$$

Recall that,

$$\nabla_u f(U,\bar{C}) = al(T - T^*)(u^{\otimes l-1}, I)c^{l-2} + \lambda lu,$$

we have

$$\left\|\nabla_u f(U,\bar{C})\right\|_F \le l\sqrt{20}\max\left(\|u\|, 2\sqrt{m+K}\delta(\sqrt{d(m+K)})^{l-2}\right) + \lambda l\|u\|$$

$$\le l\sqrt{20}\max\left(M_2, 2\sqrt{m+K}\delta(\sqrt{d(m+K)})^{l-2}\right) + \lambda l M_2$$

$$\le l\sqrt{20}M_2 + \lambda l M_2,$$

where the last inequality assumes $\delta \le \frac{M_2}{2\sqrt{m+K}(\sqrt{d(m+K)})^{l-2}}$.

Finally, we have

$$g(u') - g(u) \le \left\langle\nabla g(u), -\eta\nabla_u f(U,\bar{C})\right\rangle + \frac{\eta^2}{2}2l^2\sqrt{20}\frac{4d}{\mu_3^2\delta^2}\left\|\nabla_u f(U,\bar{C})\right\|^2$$

$$\le \frac{10l^2 2^l}{\mu_3}\eta d^{l-1.5}\sqrt{m}(m+K)^{l-1}\delta M_2 + \frac{\eta^2}{2}2l^2\sqrt{20}\frac{4d}{\mu_3^2\delta^2}\left(l\sqrt{20}M_2 + \lambda l M_2\right)^2$$

$$\le \frac{10l^2 2^l}{\mu_3}\eta d^{l-1.5}\sqrt{m}(m+K)^{l-1}\sqrt{\frac{10}{\lambda}}\delta + \sqrt{20}\eta^2 l^2\frac{4d}{\mu_3^2\delta^2}\left(40l^2\frac{10}{\lambda} + 2\lambda^2 l^2\frac{10}{\lambda}\right)$$

$$\le \mu l^4 2^l d^{l-1.5}m^{1/2}(m+K)^{l-1}\eta\delta\lambda,$$

where the last inequality assumes $l \geq 3$, $\eta \leq \delta^3$ and $\mu$ is some constant. $\qquad\square$

Therefore, the only way to change the residual term by a lot must be changing the tensor $T$, and the accumulated change of $T$ is strongly correlated with the decrease of $f$. This is similar to the technique of bounding the function value decrease in Wei et al. (2019). The connection between them are formalized in the following lemma, which is the formal version of Lemma 9:

**Lemma 21.** *Assume that* $\delta \leq \frac{\mu_1}{m^{\frac{3}{4}}\sqrt{\lambda}d^{\frac{l-2}{2}}(m+K)^{\frac{l-1}{2}}}, \eta \leq \frac{\mu_2\lambda}{m^{\frac{1}{2}}l^4 d^{\frac{l-1}{2}}(m+K)^{\frac{l-2}{2}}}$ *for some constants* $\mu_1, \mu_2,$ *and* $\frac{10}{m} \leq \lambda \leq 1$. *Within one epoch, let $T_0$ be the tensor after reinitialization, and let $T_t$ be the tensor at the end of the $t$-th iteration. Let $(U_0, \bar{C}_0)$ be the parameters after the reinitialization step and let $(U_H, \bar{C}_H)$ be the parameters at the end of this epoch. We have*

$$\sum_{\tau=1}^{H} \|T_\tau - T_{\tau-1}\|_F$$
$$\leq 200 l^{2.5}\sqrt{\frac{1}{\lambda}}\sqrt{\eta H}\sqrt{f(U_0, \bar{C}_0) - f(U_H, \bar{C}_H) + 160m(m+K)\delta^2(\sqrt{d(m+K)})^{l-2}}$$
$$+ 16m(m+K)\delta^2(\sqrt{d(m+K)})^{l-2}.$$

Intuitively, if we are doing a standard gradient descent, at each step the change in function value is going to be proportional to the square of the change in the tensor $T$, and the guarantee similar to the Lemma above can be proved by applying Cauchy-Schwartz. However, in our setting the proof becomes more complicated because of the normalization steps and in particular the scalar mode switch.

Before proving Lemma 21, we first prove the following lemma which guarantees the function value decrease in one step (without scalar mode switch):

**Lemma 22.** *Assume* $\delta \leq \frac{\mu_1}{m^{\frac{3}{4}}\sqrt{\lambda}d^{\frac{l-2}{2}}(m+K)^{\frac{l-1}{2}}}, \eta \leq \frac{\mu_2\lambda}{m^{\frac{1}{2}}l^4 d^{\frac{l-1}{2}}(m+K)^{\frac{l-2}{2}}}$ *for some constants* $\mu_1, \mu_2,$ *and* $\eta \leq \delta^3$. *Assume* $K \leq \frac{\lambda m}{14}$. *Starting from $T$ parameterized by $(U, \bar{C})$, suppose after one iteration (before potential scalar mode switch) the tensor becomes $T'$ parameterized by $(U', \bar{C}')$. We have*

$$\|T' - T\|_F \leq 200 l^2 \sqrt{\frac{1}{\lambda}}\left\|\eta\nabla_U f(U, \bar{C})\right\|_F.$$

**Proof of Lemma 22.** According to the algorithm, we know each iteration is composed of two steps: update $U$ by gradient descent ($U' = U - \eta\nabla_U f(U, \bar{C})$) and update $C$ and $\hat{C}$ according to $U'$. Let $\hat{T}$ be the intermediate tensor parameterized by $(U', \bar{C})$. We will bound $\|T' - T\|_F$ by bounding $\left\|\hat{T} - T\right\|_F$ and $\left\|T' - \hat{T}\right\|_F$ separately.

According to Lemma 16, we know $\sum_{i=1}^{m}\|u_i\|^2, \sum_{i=1}^{m}\|u_i'\|^2 \leq \frac{10}{\lambda}$. Denote $M_2^2 = \frac{10}{\lambda}$.

**Bounding** $\left\|\hat{T} - T\right\|_F$: From $T$ to $\hat{T}$, $U$ is updated to $U' = U - \eta\nabla_U f(U, \bar{C})$ while $C$ and $\hat{C}$ remains the same. Therefore,

$$\left\|\hat{T} - T\right\|_F = \left\|\sum_{i=1}^{m} a_i c_i^{l-2}\left(u_i - \eta\nabla_{u_i} f(U, \bar{C})\right)^{\otimes l} - \sum_{i=1}^{m} a_i c_i^{l-2} u_i^{\otimes l}\right\|_F$$
$$\leq \sum_{i=1}^{m} l\|u_i\|^{l-1}\left\|\eta\nabla_{u_i} f(U, \bar{C})\right\| c_i^{l-2} + \sum_{i=1}^{m}\sum_{k=2}^{l}\binom{l}{k}\|u_i\|^{l-k}\left\|\eta\nabla_{u_i} f(U, \bar{C})\right\|^k c_i^{l-2}.$$

We can further bound the linear term as follows:

$$\sum_{i=1}^{m} l\left\|u_i\right\|^{l-1}\left\|\eta\nabla_{u_i}f(U,\bar{C})\right\|c_i^{l-2} \leq l\sum_{i=1}^{m}\left\|\eta\nabla_{u_i}f(U,\bar{C})\right\|\max(\left\|u_i\right\|,2\sqrt{m+K}\delta(\sqrt{d(m+K)})^{l-2})$$

$$\leq l\sqrt{\sum_{i=1}^{m}\left\|\eta\nabla_{u_i}f(U,\bar{C})\right\|^2}\sqrt{\sum_{i=1}^{m}\max(\left\|u_i\right\|^2,4(m+K)^{l-1}\delta^2 d^{l-2})}$$

$$\leq \sqrt{2}lM_2\eta\left\|\nabla_U f(U,\bar{C})\right\|_F,$$

where the last inequality assumes $\delta^2 \leq \frac{M_2^2}{4m(m+K)^{l-1}d^{l-2}}$.

According to the proof in Lemma 14, we know $\left\|\eta\nabla_{u_i}f(U,\bar{C})\right\| \leq \frac{1}{l}\left\|u_i\right\|$. Therefore, for the higher order terms, for each $k \geq 2$,

$$\sum_{i=1}^{m}\binom{l}{k}\left\|u_i\right\|^{l-k}\left\|\eta\nabla_{u_i}f(U,\bar{C})\right\|^k c_i^{l-2} \leq \sum_{i=1}^{m}l^k\left\|u_i\right\|^{l-k}\frac{\left\|u_i\right\|^{k-1}}{l^{k-1}}\left\|\eta\nabla_{u_i}f(U,\bar{C})\right\|c_i^{l-2}$$

$$\leq \sum_{i=1}^{m}l\left\|u_i\right\|^{l-1}\left\|\eta\nabla_{u_i}f(U,\bar{C})\right\|c_i^{l-2}$$

$$\leq \sqrt{2}lM_2\eta\left\|\nabla_U f(U,\bar{C})\right\|_F.$$

Overall, we have

$$\left\|\hat{T}-T\right\|_F \leq 2\sqrt{2}l^2 M_2\eta\left\|\nabla_U f(U,\bar{C})\right\|_F.$$

**Bounding** $\left\|T'-\hat{T}\right\|_F$: From $\hat{T}$ to $T'$, we update $C$ to $C'$ and $\hat{C}$ to $\hat{C}'$ such that $\forall i \in [m], c_i' = c_i\frac{\left\|u_i\right\|}{\left\|u_i'\right\|}$ and $\hat{c}_i' = \hat{c}_i\frac{\left\|u_i\right\|}{\left\|u_i'\right\|}$. Thus,

$$\left\|T'-\hat{T}\right\|_F = \left\|\sum_{i=1}^{m}a_i(c_i')^{l-2}(u_i')^{\otimes l}-\sum_{i=1}^{m}a_i c_i^{l-2}(u_i')^{\otimes l}\right\|_F$$

$$\leq \sum_{i=1}^{m}\left|(c_i')^{l-2}-c_i^{l-2}\right|\left\|u_i'\right\|^l.$$

Now, let's focus on the change in $c_i^{l-2}$. Define $g(u) = \frac{1}{\left\|u\right\|^{l-2}}$. We have,

$$\nabla g(u) = -(l-2)\frac{u}{\left\|u\right\|^l} \text{ and } \nabla^2 g(u) = -(l-2)\frac{I}{\left\|u\right\|^l}+l(l-2)\frac{uu^\top}{\left\|u\right\|^{l+2}}.$$

Therefore, the spectral norm of $\nabla^2 g(u)$ is bounded by $l^2/\left\|u\right\|^l$.

For any $i \in [m]$, let $u_i''$ be any point on the line segment between $u_i$ and $u_i'$, then $\left\|\nabla^2 g(u_i'')\right\|_2 \leq l^2/\left\|u_i''\right\|^l$. If $c_i = 1/\left\|u_i\right\|$, we have

$$\left|(c_i')^{l-2}-c_i^{l-2}\right| \leq \left\|\nabla g(u_i)\right\|\left\|\eta\nabla_{u_i}f(U,\bar{C})\right\|+\frac{1}{2}\max_{u_i''}\left\|\nabla^2 g(u_i'')\right\|_2\left\|\eta\nabla_{u_i}f(U,\bar{C})\right\|^2$$

$$\leq \frac{l-2}{\left\|u_i\right\|^{l-1}}\left\|\eta\nabla_{u_i}f(U,\bar{C})\right\|+\frac{1}{2}\max_{u_i''}\frac{l\left\|u_i\right\|}{\left\|u_i''\right\|^l}\left\|\eta\nabla_{u_i}f(U,\bar{C})\right\|.$$

If $c_i = \sqrt{d(m+K)}/\left\|u_i\right\|$, we have

$$\left|(c_i')^{l-2}-c_i^{l-2}\right| \leq \frac{l-2}{\left\|u_i\right\|^{l-1}}\left\|\eta\nabla_{u_i}f(U,\bar{C})\right\|(\sqrt{d(m+K)})^{l-2}+\frac{1}{2}\max_{u_i''}\frac{l\left\|u_i\right\|}{\left\|u_i''\right\|^l}\left\|\eta\nabla_{u_i}f(U,\bar{C})\right\|(\sqrt{d(m+K)})^{l-2}.$$

Therefore, we have

$$\left\| T' - \hat{T} \right\|_F \leq 2el \sum_{i=1}^{m} \left\| \eta \nabla_{u_i} f(U, \bar{C}) \right\| \max(\|u_i\|, 2\sqrt{m+K}\delta(\sqrt{d(m+K)})^{l-2})$$

$$\leq 2\sqrt{2}elM_2 \left\| \eta \nabla_U f(U, \bar{C}) \right\|_F,$$

where the first inequality holds because $\|u_i'\| \leq \left(1 + \frac{1}{l}\right)\|u_i\|$ and the second inequality assumes $\delta^2 \leq \frac{M_2^2}{4m(m+K)^{l-1}d^{l-2}}$.

Overall, combing the bounds on $\left\| \hat{T} - T \right\|_F$ and $\left\| T' - \hat{T} \right\|_F$, we have

$$\|T' - T\|_F \leq 200l^2 \sqrt{\frac{1}{\lambda}} \left\| \eta \nabla_U f(U, \bar{C}) \right\|_F.$$

$\square$

Now we are ready to prove Lemma 21.

**Proof of Lemma 21.** Let's first bound the tensor change and function value change due to scalar mode switches. Following the proof of Claim 2 in Lemma 15, setting $\tilde{\Gamma} = 10$ and assuming $\eta \leq \frac{1}{l(\sqrt{2\tilde{\Gamma}}(\sqrt{d(m+K)})^{l-2}+\lambda)}$, we know each scalar mode switch can at most change the tensor Frobenius norm by $16(m+K)\delta^2(\sqrt{d(m+K)})^{l-2}$. Furthermore, using the same argument as Claim 2, the function value can increase by at most $\sqrt{20}\left(16(m+K)\delta^2(\sqrt{d(m+K)})^{l-2}\right) + \frac{1}{2}\left(16(m+K)\delta^2(\sqrt{d(m+K)})^{l-2}\right)^2 \leq 160(m+K)\delta^2(\sqrt{d(m+K)})^{l-2}$, where we assume $\delta^2 \leq \frac{5}{8(m+K)(\sqrt{d(m+K)})^{l-2}}$.

According to the algorithm, we know each epoch contains at most $m$ scalar mode switches. Suppose $T'_\tau$ be the tensor before potential scalar mode switch in the $\tau$-th iteration. Then, we have

$$\sum_{\tau=1}^{t} \|T_\tau - T_{\tau-1}\|_F \leq \sum_{\tau=1}^{t} \|T'_\tau - T_{\tau-1}\|_F + \sum_{\tau=1}^{t} \|T_\tau - T'_\tau\|_F$$

$$\leq \sum_{\tau=1}^{t} \|T'_\tau - T_{\tau-1}\|_F + 16m(m+K)\delta^2(\sqrt{d(m+K)})^{l-2}.$$

According to Lemma 22, we know

$$\|T'_\tau - T_{\tau-1}\|_F \leq 200l^2 \sqrt{\frac{1}{\lambda}} \left\| \eta \nabla_U f(U_{\tau-1}, \bar{C}_{\tau-1}) \right\|_F.$$

Therefore, we have

$$\sum_{\tau=1}^{t} \|T'_\tau - T_{\tau-1}\|_F \leq 200l^2 \sqrt{\frac{1}{\lambda}} \sum_{\tau=1}^{t} \left\| \eta \nabla_U f(U_{\tau-1}, \bar{C}_{\tau-1}) \right\|_F$$

$$\leq 200l^2 \sqrt{\frac{1}{\lambda}} \sqrt{t} \sqrt{\sum_{\tau=1}^{t} \left\| \eta \nabla_U f(U_{\tau-1}, \bar{C}_{\tau-1}) \right\|_F^2}.$$

According to Lemma 14, we know $f(U'_\tau, \bar{C}'_\tau) - f(U_{\tau-1}, \bar{C}_{\tau-1}) \leq -\frac{\eta}{l} \left\| \nabla_U f(U_{\tau-1}, \bar{C}_{\tau-1}) \right\|_F^2$. Therefore, we have

$$\sum_{\tau=1}^{t} \|T'_\tau - T_{\tau-1}\|_F \leq 200l^2 \sqrt{\frac{1}{\lambda}} \sqrt{t} \sqrt{\sum_{\tau=1}^{t} \eta l \left( f(U_{\tau-1}, \bar{C}_{\tau-1}) - f(U'_\tau, \bar{C}'_\tau) \right)}.$$

Since scalar mode switches in total change the function value by at most $160m(m + K)\delta^2(\sqrt{d(m + K)})^{l-2}$, we know

$$\sum_{\tau=1}^{t} \left( f(U_{\tau-1}, \bar{C}_{\tau-1}) - f(U'_\tau, \bar{C}'_\tau) \right)$$
$$\leq f(U_0, \bar{C}_0) - f(U_t, \bar{C}_t) + 160m(m + K)\delta^2(\sqrt{d(m + K)})^{l-2}.$$

Overall, we have

$$\sum_{\tau=1}^{t} \|T_\tau - T_{\tau-1}\|_F$$
$$\leq 200l^{2.5}\sqrt{\frac{1}{\lambda}}\sqrt{\eta H}\sqrt{f(U_0, \bar{C}_0) - f(U_t, \bar{C}_t) + 160m(m + K)\delta^2(\sqrt{d(m + K)})^{l-2}}$$
$$+ 16m(m + K)\delta^2(\sqrt{d(m + K)})^{l-2}.$$

$\square$

Combining all the steps above, we are now ready to prove Lemma 12.

**Proof of Lemma 12.** Let $u_0$ be the reinitialized vector. According to Lemma 18, we know with probability at least $1/5$,

$$a(P_{S^{\otimes l}}T'_0 - P_{S^{\otimes l}}T^*)(\overline{P_S u_0}^{\otimes l}) \leq \frac{-1}{(\mu_1 rl)^{l/2}} \|P_{S^{\otimes l}}T'_0 - P_{S^{\otimes l}}T^*\|_F \leq -\frac{\epsilon}{(\mu_1 rl)^{l/2}},$$

where $\mu_1$ is some constant. According to Lemma 23, we know with probability at least $1 - 1/30$, $\|P_S u_0\| \geq \frac{\mu_2 \delta}{\sqrt{d}}$ for some constant $\mu_2 < 1$. Taking a union bound, we know both properties hold with probability at least $1/6$.

Conditioning on both properties, we will prove that

$$f(U_0, \bar{C}_0) - f(U_H, \bar{C}_H) \geq \frac{\lambda}{32000000(\mu_1 rl)^l \eta H l^5} \epsilon^2.$$

For the sake of contradiction, assume that $f(U_0, \bar{C}_0) - f(U_H, \bar{C}_H) \leq \frac{\lambda}{32000000(\mu_1 rl)^l \eta H l^5} \epsilon^2$. According to Lemma 21, we know

$$\sum_{\tau=1}^{H} \|T_\tau - T_{\tau-1}\|_F \leq \frac{\epsilon}{10(\mu_1 rl)^{l/2}}$$

as long as $\delta^2 \leq \frac{\epsilon}{320(\mu_1 rl)^{l/2}m(m+K)^{\frac{l}{2}}d^{\frac{l-2}{2}}}$ and $\delta^2 \leq \frac{\lambda\epsilon^2}{32000000(\mu_1 rl)^l \eta H l^5 \cdot 160m(m+K)^{\frac{l}{2}}d^{\frac{l-2}{2}}}$.

We will prove that $a(P_{S^{\otimes l}}T_t - P_{S^{\otimes l}}T^*)(\overline{P_S u_t}^{\otimes l}) \leq -\frac{\epsilon}{5(C_1 rl)^{l/2}}$ for all $0 \leq t \leq H - 1$, so from Lemma 19 we know that the norm of $P_S u_t$ must increase exponentially.

Let's first prove the case at the beginning of an epoch: Let $T_0$ be the tensor after reinitialization. According to the proof of Claim 1 in Lemma 15, we know

$$\|T_0 - T'_0\|_F \leq 2\sqrt{\frac{10}{\lambda m}} \leq \frac{\epsilon}{2(\mu_1 rl)^{l/2}},$$

where the last inequality assumes $\lambda m \geq \frac{160(\mu_1 rl)^l}{\epsilon^2}$. This implies that

$$a(P_{S^{\otimes l}}T_0 - P_{S^{\otimes l}}T^*)(\overline{P_S u_0}^{\otimes l}) \leq a(P_{S^{\otimes l}}T'_0 - P_{S^{\otimes l}}T^*)(\overline{P_S u_0}^{\otimes l}) + \|T_0 - T'_0\|_F \leq -\frac{\epsilon}{2(\mu_1 rl)^{l/2}}.$$

For later steps, we will show that $a(P_{S^{\otimes l}}T_t - P_{S^{\otimes l}}T^*)(\overline{P_S u_t}^{\otimes l})$ is close to $a(P_{S^{\otimes l}}T_0 - P_{S^{\otimes l}}T^*)(\overline{P_S u_0}^{\otimes l})$. Actually,

$$\left| a(P_{S^{\otimes l}}T_t - P_{S^{\otimes l}}T^*)(\overline{P_S u_t}^{\otimes l}) - a(P_{S^{\otimes l}}T_0 - P_{S^{\otimes l}}T^*)(\overline{P_S u_0}^{\otimes l}) \right|$$

$$\leq \left| \sum_{\tau=1}^{t} \left( (P_{S^{\otimes l}}T_{\tau-1} - P_{S^{\otimes l}}T^*)(\overline{P_S u_\tau}^{\otimes l}) - (P_{S^{\otimes l}}T_{\tau-1} - P_{S^{\otimes l}}T^*)(\overline{P_S u_{\tau-1}}^{\otimes l}) \right) \right|$$

$$+ \left| \sum_{\tau=1}^{t} \left( (P_{S^{\otimes l}}T_\tau - P_{S^{\otimes l}}T^*)(\overline{P_S u_\tau}^{\otimes l}) - (P_{S^{\otimes l}}T_{\tau-1} - P_{S^{\otimes l}}T^*)(\overline{P_S u_\tau}^{\otimes l}) \right) \right|$$

$$\leq H\mu l^4 2^l d^{l-1.5} m^{1/2}(m+K)^{l-1}\eta\delta\lambda + \sum_{\tau=1}^{t} \|T_\tau - T_{\tau-1}\|$$

$$\leq H\mu l^4 2^l d^{l-1.5} m^{1/2}(m+K)^{l-1}\eta\delta\lambda + \frac{\epsilon}{10(\mu_1 rl)^{l/2}} \leq \frac{\epsilon}{5(\mu_1 rl)^{l/2}}.$$

The second inequality above comes from Lemma 20, and the last inequality assumes $\delta \leq \frac{1}{\mu l^4 2^l d^{l-1.5} m^{1/2}(m+K)^{l-1}\eta\lambda H} \cdot \frac{\epsilon}{10(\mu_1 rl)^{l/2}}$.

This then implies that for all $0 \leq t \leq H-1$,

$$a(P_{S^{\otimes l}}T_t - P_{S^{\otimes l}}T^*)(\overline{P_S u_t}^{\otimes l}) \leq -\frac{\epsilon}{2(\mu_1 rl)^{l/2}} + \frac{\epsilon}{5(\mu_1 rl)^{l/2}} \leq -\frac{\epsilon}{5(\mu_1 rl)^{l/2}}.$$

Then according to Lemma 19,

$$\|P_S u_H\|^2 \geq \left( 1 + \eta \left(\frac{\mu_2}{2}\right)^{l-2} \frac{\epsilon}{10(\mu_1 rl)^{l/2}} \right)^H \|P_S u_0\|^2$$

$$\geq \left( 1 + \eta \left(\frac{\mu_2}{2}\right)^{l-2} \frac{\epsilon}{10(\mu_1 rl)^{l/2}} \right)^H \frac{\mu_2^2 \delta^2}{d}$$

$$\geq \exp\left( \frac{1}{2}\eta H \left(\frac{\mu_2}{2}\right)^{l-2} \frac{\epsilon}{10(\mu_1 rl)^{l/2}} \right) \frac{\mu_2^2 \delta^2}{d},$$

where the last inequality assumes $\eta \leq \left(\frac{2}{\mu_2}\right)^{l-2} \frac{10(\mu_1 rl)^{l/2}}{\epsilon}$. Therefore, $\|P_S u_H\|^2$ exceeds $M_2$ as long as $\eta H \geq 2 \left(\frac{2}{\mu_2}\right)^{l-2} \frac{10(\mu_1 rl)^{l/2}}{\epsilon} \log\left(\frac{dM_2}{\mu_2^2 \delta^2}\right)$. Since $M_2 = \sqrt{\frac{10}{\lambda}}$ is the upper bound of $\|U\|_F$, this finishes the contradiction proof.

We have shown that

$$f(U_0, \bar{C}_0) - f(U_H, \bar{C}_H) \geq \frac{\lambda}{32000000(\mu_1 rl)^l \eta H l^5} \epsilon^2.$$

In order to show $f(U_0', \bar{C}_0') - f(U_H, \bar{C}_H)$ is large, we still need to bound $|f(U_0', \bar{C}_0') - f(U_0, \bar{C}_0)|$ that comes from reinitialization. According to Lemma 15, we know

$$|f(U_0', \bar{C}_0') - f(U_0, \bar{C}_0)| \leq \frac{200}{\lambda m} \leq \frac{\lambda}{64000000(\mu_1 rl)^l \eta H l^5} \epsilon^2,$$

where the second inequality assumes $\lambda^2 m \geq 1.28 \times 10^{11}(\mu_1 rl)^l \eta H l^5$. Therefore,

$$f(U_H, \bar{C}_H) - f(U_0', \bar{C}_0')$$

$$\leq \left( f(U_0, \bar{C}_0) - f(U_H, \bar{C}_H) \right) + |f(U_0, \bar{C}_0) - f(U_0', \bar{C}_0')|$$

$$\leq -\frac{\lambda}{3.2 \times 10^7(\mu_1 rl)^l \eta H l^5}\epsilon^2 + \frac{\lambda}{6.4 \times 10^8(\mu_1 rl)^l \eta H l^5}\epsilon^2$$

$$\leq -\frac{\lambda}{6.4 \times 10^7(\mu_1 rl)^l \eta H l^5}\epsilon^2.$$

We choose $m = O\left(\frac{r^{2.5l}}{\epsilon^5}\log(d/\epsilon)\right)$, $\lambda = O\left(\frac{\epsilon}{r^{0.5l}}\right)$, $\delta = O\left(\frac{\epsilon^{5l-1.5}}{d^{l-1.5}(\log(d/\epsilon))^{l+0.5}r^{2.5l^2-0.75l}}\right)$, $\eta = O\left(\frac{\epsilon^{15l-4.5}}{d^{3l-4.5}(\log(d/\epsilon))^{3l+1.5}r^{7.5l^2-2.25l}}\right)$, $H = O\left(\frac{d^{3l-4.5}(\log(d/\epsilon))^{3l+2.5}r^{7.5l^2-1.75l}}{\epsilon^{15l-3.5}}\right)$ and $K = O\left(\frac{r^{2l}}{\epsilon^4}\log(d/\epsilon)\right)$ such that all the conditions are satisfied and the function value decreases by $\Omega(\frac{\epsilon^4}{r^{2l}\log(d/\epsilon)})$ in each epoch. Note that there does exist some circular dependency between the parameters. This turns out to be not an issue in our proof because for example $\delta$ depends on $\frac{1}{\eta H}$ while $\eta H$ only depends logarithmically on $1/\delta$. Other circular dependencies can be resolved in the same manner. $\qquad\square$

## D  Tools

In this section, we give the technical lemmas we use in the proof.

### D.1  Random projection on a subspace

We use the following lemma to show that with good probability, the projection of the reinitialized component on the good subspace is lower bounded.

**Lemma 23** (Lemma 2.2 in Dasgupta and Gupta (2003)). *Let $Y$ be a $d$-dimensional vector uniformly sampled from sphere $\mathbb{S}^{d-1}$. Let $Z \in \mathbb{R}^k$ be the projection of $Y$ onto its first $k$ coordinates ($k < d$). For any $\beta < 1$, we have*

$$\Pr\left[\|Z\|^2 \le \frac{\beta k}{d}\right] \le \exp\left(\frac{k}{2}(1 - \beta + \ln\beta)\right).$$

### D.2  Norm of random Gaussian vectors

The following lemma gives the concentration of $\ell_2$ norm of a random Gaussian vector.

**Lemma 24** (Theorem 3.1.1 in Vershynin (2018)). *Let $X = (X_1, X_2, \cdots, X_n) \in \mathbb{R}^n$ be a random vector with each entry independently sampled from $\mathcal{N}(0,1)$. Then*

$$\Pr\left[\big|\|x\| - \sqrt{n}\big| \ge t\right] \le 2\exp\left(-t^2/C^2\right),$$

*where $C$ is an absolute constant.*

### D.3  Anti-concentration of Gaussian polynomials

We use anti-concentration of Gaussian polynomials to argue that a randomly initialized component has good correlation with the residual.

**Lemma 25** (Theorem 8 in Carbery and Wright (2001)). *Let $x \in \mathbb{R}^n$ be a Gaussian variable $x \in N(0, I)$, for any polynomial $p(x)$ of degree $l$, there exists a constant $\kappa$ such that*

$$\Pr\left[|p(x)| \le \epsilon\sqrt{Var[p(x)]}\right] \le \kappa l \epsilon^{1/l}.$$