[Reviews · NeurIPS 2020]

Review 1

Summary and Contributions: The paper focuses on an over-parameterized tensor decomposition that is closely connected to over-parameterized neural networks and hence of interest to the NeurIPS community.

Strengths: Very strong theoretical result, probably worthy of a leading theory conference as well (SODA comes to mind, more than FOCS/STOC). Tensor decompositions are NP-hard (when the rank is unspecified). Assuming that the target rank is r (a priori know and fixed) the authors present a variant of a gradient descent algorithm that finds a decomposition with m components (informally speaking, rank-m) with m much, much smaller than prior work (depends on r^\ell as opposed to d^\ell, using the notation of the paper, d and \ell are the tensor dimensions and number of modes). This is a major and non-trivial improvement.

Weaknesses: No experimental component, but I would not count this as a weakness. This is a theory paper and NeurIPS should accept pure theory papers as well.

Correctness: Did not check. There are 40 pages of proofs that I cannot parse in the very short review period. The proof outline makes sense, but I did not check the details.

Clarity: Yes.

Relation to Prior Work: I am not the foremost expert in this area, but many relevant papers have been discussed.

Reproducibility: Yes

Additional Feedback: I had no real comments for the authors and I feel that their response addressed concerns raised by the other reviewers.


Review 2

Summary and Contributions: This paper addressed the problem of how to find a better over-parametrized rank of tensor decomposition. A variant of the gradient descent algorithm for tensor decomposition is proposed and plentiful theoretical results were provided to prove the proposed algorithm can approximate a tensor by a lower rank bound than the lazy training regime.

Strengths: Theoretically proving the existence of a lower approximation rank can be acquired by a novel gradient descent method is interesting and the results are promising. The proposed gradient descent method and theoretical contributions are potentially useful in increasing the performance of neural network training.

Weaknesses: Though strong theoretical proofs were given in the paper, the effectiveness of the proposed gradient descent method in neural network training still remains questionable. It is better to provide some numerical experiments to support the theoretical results of the lower bound and the improvement of the neural network training when applying the proposed gradient descent method.

Correctness: The theoretical results in this paper are plentiful and look sound.

Clarity: The paper is well-structured and the references are properly cited. The paper can explain the motivation and the proposed method well. Minor typos: Below Line 85, quote 1: effecitively --> effectively Line 165: min --> minimum Line 210: high level --> high-level Line 316: gradient based --> gradient-based

Relation to Prior Work: The paper addressed the related work of the lazy training regime and the existing problems and challenges of over-parametrization. The paper provided the clear comparison between the previous work and the proposed results.

Reproducibility: No

Additional Feedback:


Review 3

Summary and Contributions: This paper consider tensor factorization problem that decomposes a tensor into a set of rank-1 tensors. The main contribution of this paper is to propose a nonconvex optimization algorithm and to show it provably converges to a global minimum under over-parameterization setting, i.e., m = O(r^{2.5 \ell}), where m is the number of components in the optimization variable, r is the rank of the target tensor, and \ell is the order of the tensor.

Strengths: 1. This paper proposes a reparameterized nonconvex optimization formulation for tensor decomposition to avoid higher order saddles. 2. The authors then proposed a gradient descent based optimization algorithm and proved it converges to a global minimum with high probability.

Weaknesses: 1. The main concern is about the utilization of the over-parameterization in tensor decomposition. In other words, for a rank-r tensor, tensor decomposition aims to find the r components which have physical interpretation. However, the proposed approach instead finds m = O(r^{2.5 \ell}) components, which could be far away from the target r components. For example, even for third-order tensor, it finds m = O(r^7.5) components, much larger than r. 2. Perhaps the goal of this paper is to understand the effect of over-parameterization in tensor decomposition. However, if this is the case, the objective function is quite different to the classical one, and the algorithm is also different to simple gradient descent. The algorithm should also be numerically compared with existing algorithms.

Correctness: They are correct, but I didn't carefully check all the proofs in Appendix.

Clarity: Overall, the paper is well written and easy to follow. In Thm 3, does it require r < d or not? I guess there is no restriction on r, but it needs to be stated to make it clear, since in other place like Thm 2 it requires r<d. There are few grammar issue, e.g., line 20-21: we consider..., we aim

Relation to Prior Work: Relation to prior work is clearly discussed.

Reproducibility: Yes

Additional Feedback: ----------------------------------After rebuttal--------------------------------------------- I am still not sure how practical the current results are, but I think it will be more interesting if the results can be extended to other tasks such as tensor completion.


Review 4

Summary and Contributions: Consider a low-rank tensor decomposition problem: given a rank-r tensor T, find a rank-m tensor that approximates T with small l2 error. If m is as large as the number of elements in T, we can find a global optimum, but the setting is impractical. In this paper, a gradient descent algorithm along with a new tensor parameterization is proposed, which is theoretically guaranteed to minimize the l2 error arbitrary small with much smaller m.

Strengths: - The paper is clearly written and easy to read - Theoretically clear results - Interesting research direction that tries to connect tensor decomposition and neural tangent kernel

Weaknesses: - No empirical verification - Presented results are not ready to show the direct connection to neural tangent kernel (and the learning dynamics of neural networks)

Correctness: The obtained results seem correct (I did sanity check of the results but I didn't check the proofs)

Clarity: The paper is very clearly written.

Relation to Prior Work: Related work is sufficintly referenced.

Reproducibility: Yes

Additional Feedback: This is an interesting study that tries to go beyond the previous results in tensor decomposition methods and make some bridge to the learning dynamics of neural networks. Although I feel the bridging part is under development, the main contributions are still significant. To improve the quality of the paper, I have two suggestions. 1. It would be better to explain some intuitions for the new parameterization (the top equation in p5). I guess c_i is the normalizing term, but a_i looks mysterious to me. Could you explain why do we need a_i (or what would happen without a_i)? 2. It would be more convincing to have numerical results (e.g., comparing the error curve between normal gradient descent and the proposed algorithm).

[Author Response · NeurIPS 2020]

We thank all the reviewers for their thorough feedback and valuable suggestions! We will revise our paper accordingly.

**To Reviewer 1:**

Thanks for the positive review!

**To Reviewer 2:**

*"Effectiveness of proposed algorithm in training neural networks":*

The goal of our paper is not to propose a new algorithm that outperforms current ones in training neural networks, but
rather to analyze gradient descent on tensor decompositions beyond the lazy training regime. Tensor decomposition
problems are closely related to the training of neural networks, e.g., the population loss of one-hidden-layer networks
is a sum of tensor decompositions (Ge et al., 2017), but our algorithm cannot be directly applied to neural network
training. For the tensor decomposition problem stated in our paper, our modifications to the vanilla objective and the
vanilla GD are mostly motivated by theoretical challenges: reparameterize the objective to avoid high-order saddle
points; re-initialize one component to escape bad local minimum; etc. Some of the changes, e.g., re-initialization of
components, are extendable to neural network training, while others are more restricted to tensor decompositions.

*"Numerical experiments":*
We will add numerical experiments to verify our lower bound for lazy training on tensor decomposition problems
(Theorem 1). We modified vanilla GD mostly because of the theoretical challenges in analyzing the optimization of
tensor decompositions. We do not claim our algorithm outperforms SGD/Adam in training neural networks.

We will also fix the typos. Thanks for pointing them out.

**To Reviewer 3:**

*"Concern on the over-parameterization:"*
Recovering a rank $r$ tensor using exactly $r$ components is NP-Hard, so it's natural to use more components to fit
the ground truth tensor. Besides, in this paper, instead of optimizing the degree of over-parameterization, we focus
on studying the optimization of GD in over-parameterized tensor decompositions beyond the lazy training regime.
Overparameterization plays an essential rule in the training of neural networks; it's also known that the training of
networks can be cast as mixture of tensor decompositions (Ge et al., 2017). Therefore, we view this work as a first step
towards understanding the training of over-parameterized neural networks.

*"Non-standard objective function and optimization algorithm":*
We modified the standard objective function and vanilla GD to overcome challenges in theoretical analysis. Most
of these changes are well justified: reparameterize the objective to avoid high order saddle points; re-initialize one
component to escape bad local minimum; regularize the objective to control parameter norms. Some others might be
artifacts of our analysis: update separately on $U$ and $C, \hat{C}$; switch the scalar mode when a component grows large.
Proving similar guarantees on a more standard objective and a cleaner algorithm is an important future direction.

*"Theorem 3 requires $r < d$?":*
Theorem 3 holds for $r > d$ if we replace $r$ by $d$ in the bounds of $m, \lambda$, and $K$. However, in this setting, there are no
benefit of using this approach compared to lazy training.

We will also fix the grammar issues. Thanks for pointing that out.

**To Reviewer 4:**

*"Why do we need $a_i$?":*
Each $a_i$ is initialized as $+1$ or $-1$ and then fixed throughout the training. We need positive and negative $a_i$'s so that our
model can fit a "non-positive-definite" ground truth tensor, particularly when the order $l$ is even. For example, if the
ground truth tensor is $-v^{\otimes 4}$ for some vector $v$, our model cannot fit it if all $a_i$'s are $+1$.

*"Numerical experiments: compare normal GD and the proposed algorithm":*
We modified the normal GD to overcome challenges in theoretical analysis: re-initialize one component to escape bad
local minimum; update separately on $U$ and $C, \hat{C}$ to contract $B$ subspace. In the numerical experiments, from a random
initialization, normal GD can also successfully optimize the model as the modified algorithm. However, it will require a
significantly different proof as the proof needs to show why the trajectory of gradient descent does not go through any
spurious local minima. We leave that as a future direction.

# References

Ge, R., Lee, J. D., and Ma, T. (2017). Learning one-hidden-layer neural networks with landscape design. *arXiv preprint*
*arXiv:1711.00501*.


[Meta-Review · NeurIPS 2020]

This is a good contribution, with highly non-trivial theoretical results about the role of over-parameterization in tensor decomposition. Some reviewers are worried about the lack of numerical experiments and the weak connection to practical algorithms, but this is acceptable for papers with solid theoretical contribution at NeurIPS. The authors make it clear in the rebuttal that their goal is not develop an algorithm for neural networks. They also promised to add some numerical experiments. For these reasons, I recommend accept (poster).